

**Stable carbon isotope ratios of ambient volatile organic compounds**
**Anna Kornilova[1], Lin Huang[2], Marina Saccon[1], Jochen Rudolph[1]**
[1]Centre for Atmospheric Chemistry, York University, Toronto, ON, Canada, M3J 1P3;
[2]Environment Canada, Toronto, ON, Canada, M3H 5T4
**Correspondence to: Jochen Rudolph (rudolphj@yorku.ca)**



**Abstract.** Measurements of mixing ratios and stable carbon isotope ratios of aromatic volatile
organic compounds (VOC) in the atmosphere were made in Toronto (Canada) in 2009 and 2010.
Consistent with the kinetic isotope effect for reactions of aromatic VOC with the OH-radical the
observed stable carbon isotope ratios are on average significantly heavier than the isotope ratios
of their emissions. The change of carbon isotope ratio between emission and observation is used
to determine the extent of photochemical processing (photochemical age, $\int[OH]dt$) of the
different VOC. It is found that $\int[OH]dt$ of different VOC depends strongly on the VOC
reactivity. This demonstrates that for this set of observations the assumption of a uniform
$\int[OH]dt$ for VOC with different reactivity is not justified and that the observed values for
$\int[OH]dt$ are the result of mixing of VOC from air masses with different values for $\int[OH]dt$.
Based on comparison between carbon isotope ratios and VOC concentration ratios it is also
found that varying influence of sources with different VOC emission ratios has a larger impact
on VOC concentration ratios than photochemical processing. It is concluded that for this data set
the use of VOC concentration ratios to determine $\int[OH]dt$ would result in values for $\int[OH]dt$
inconsistent with carbon isotope ratios and that the concept of a uniform $\int[OH]dt$ for an air mass
has to be replaced by the concept of individual values of an average $\int[OH]dt$ for VOC with
different reactivity.
**1. Introduction**

20         Anthropogenic volatile organic compounds (VOC) are important pollutants that play key

roles in the production of ozone, aerosol formation and significantly affect regional air quality in
general. Their total annual global emission is estimated at 150 TgC per year (Niedojadlo et al.,
2008; Piccot et al., 1992). On a global scale, about 60 to 80% of anthropogenic emissions are





associated with fossil fuel production, its distribution, use and storage, and up to 20 to 30% with
biomass burning (Reimann and Lewis, 2007; Rudolph et al., 2002). Since anthropogenic VOC
are rather diverse, classification of their emission sources is quite challenging, thus frequently
these emissions are grouped according to the commodities or activities with which they are
associated (Niedojadlo et al., 2008; Sawyer et al., 2000; Piccot et al., 1992; Watson et al., 1991),
for instance, vehicular exhaust, evaporated fuel, solvent use, etc. Aromatic VOC are an important
part of anthropogenic emissions (up to 44%), with benzene, toluene, ethylbenzene and xylenes as
major components (up to 75%) (Jang and Kamens, 2001).

9        In the atmosphere many VOC undergo chemical transformation via gas-phase reactions

with hydroxyl radicals (OH), nitrate radicals ($NO_3$), chlorine radicals (Cl) and ozone ($O_3$), with
OH contributing the most to these oxidation processes. This oxidative processing is especially
important for aromatic VOC and heavy alkanes, since it may result in the formation of
oxygenated and nitrated products that may contribute to the formation of secondary organic
matter (Saccon et al., 2015; Irei et al., 2006; Forstener et al., 1997).

15        It has been shown by many studies that the use of stable carbon isotope ratios is

beneficial in providing insights into photochemical transformation and physical processing of
ambient volatile organic compounds (Gensch et al., 2014; Rudolph, 2007; Stein and Rudolph,
2007; Ghosh and Brand, 2003; Goldstein and Shaw, 2003; Rudolph et al., 2003; Rudolph et al.,
2002; Meier-Augenstein, 1999). McKinney et al. (1950) introduced one of the first mass
spectrometers that were able to differentiate carbon and oxygen isotopologues in carbon dioxide
and oxygen molecules (Richet et al., 1977; McKinney et al., 1950). This development was
followed by extensive studies of stable carbon isotope composition of carbon monoxide, carbon
dioxide and methane in ambient samples (Brenninkmeijer et al., 1995; Lowe et al., 1994; Stevens



et al., 1972) and theoretical modelling and interpretations began to be presented (Brenninkmeijer
et al., 1995; Kaye, 1987; Richet et al., 1977; Craig, 1953).

3        While some trace gases, such as methane, carbon monoxide, or carbon dioxide are

present in the atmosphere at µmol mol$^{-1}$ or high nmol mol-$^{1}$ levels and are easy to measure with
state of the art isotope ratio mass spectrometry, most VOC are present at mixing ratios that are 3
to 5 orders of magnitude lower, thus making the measurement of their isotopic composition
challenging. In 1997 a method which allowed the determination of stable carbon isotope
composition for ambient compounds present at low mixing ratios was introduced (Rudolph et al.,
1997). It was followed by isotope ratio measurements of various VOC, where it was applied and
expanded (Kornilova et al., 2015a; Kornilova et al., 2015b; Kawashima and Murakami, 2014;
Wintel et al., 2013; Eckstaedt et al., 2012; Giebel et al., 2010; Redeker et al., 2007; Nara et al.,
2006; Turner et al., 2006; Komatsu et al., 2005; Czapiewski et al., 2002; Rudolph et al., 2002;
Saito et al., 2002). Still, the number of publications containing measurements of isotope ratios of
ambient VOC and their sources is very limited (Kornilova et al., 2015b; Gensch et al., 2014;
Elsner et al., 2012).

16       The stable carbon isotope ratio is determined by the ratio of $^{13}C$ to $^{12}C$ atoms in the

sample and is usually expressed in delta notation ($\delta^{13}C$) relative to a reference value (Vienna
Peedee Belemnite, VPDB) with $^{13}C/^{12}C = 0.0112372$ (Craig, 1953)). Since changes in $^{13}C/^{12}C$
are small, $\delta^{13}C$ is expressed in parts per thousand (‰) (Eq. 1).

$$\delta^{13}C = \frac{R_{sample}(^{13}C/^{12}C) - R_{VPDB}(^{13}C/^{12}C)}{R_{VPDB}(^{13}C/^{12}C)} \times 1000‰ \qquad \text{Eq. 1}$$



The presence of $^{13}$C isotopes slightly affects the reaction rate of the molecule, as $^{13}$C containing
isotopologues react slower than $^{12}$C-only isotopologues. The relative difference between these
rate constants ($^{12}$k and $^{13}$k), known as kinetic isotope effect (KIE), is expressed in epsilon
notation ($\varepsilon$) in parts per thousand (‰) (Eq. 2).

$$\varepsilon_{OH} = \frac{^{12}k - ^{13}k}{^{13}k} \times 1000‰$$    Eq. 2

Since $^{12}$C is more reactive, the sample gets enriched in $^{13}$C with time and the magnitude of this
depends on the extent of processing as well as the KIE.

9       The term "photochemical age" is used for quantification of photochemical processing of

a compound and it is usually defined as the time integral of the OH concentration for an air mass
($\int[OH]dt$). Originally VOC concentration ratios were used to determine $\int[OH]dt$, an approach
that often is referred to as hydrocarbon clock (Kornilova et al., 2015b; Stein and Rudolph, 2007;
Kleinman et al., 2003; Rudolph et al., 2003; Thompson et al., 2003; Rudolph and Czuba, 2000;
Jobson et al., 1999; Jobson et al., 1998). The isotope hydrocarbon clock approach is similar, but
uses stable carbon isotope ratios (Rudolph and Czuba, 2000) (Eq. 3):

$$\int[OH]dt = \frac{\delta_A{}^{13}C - \delta_S{}^{13}C}{k_{OH}\varepsilon_{OH}}$$    Eq. 3

where $\int[OH]dt$ is the average photochemical age, $\delta_A{}^{13}C$ and $\delta_A{}^{13}C$ are the stable carbon isotope
compositions of ambient and freshly emitted VOC respectively, $k_{OH}$ is the rate constant for the
reaction of VOC and OH, and $\varepsilon_{OH}$ is the KIE (Eq.2).



In this paper we present stable carbon isotope ratios of ambient aromatic VOC measured
at a suburban area in 2009 and 2010. The results will be used to determine $\int[OH]dt$ for VOC with
different reactivity and to determine the impact of photochemical processing on the change in
VOC concentration ratios.
**2. Materials and method**
Selective VOC sampling from ambient air was done using cartridges (13 to 15 cm long,
1/4" OD) filled with Carboxene 569, a mass flow controller and a diaphragm pump with a
sampling flow rate set between 10 and 50 mL min$^{-1}$ as described in detail by Kornilova et al.
(2015a). VOC were quantitatively extracted from cartridges using a thermal desorption unit with
helium as a carrier gas (60 to 80 mL min$^{-1}$) at 553 K for 40 min. Desorbed VOC were focused
using a two stage preconcentration system. Compounds were then separated using a HP5890
Series II gas chromatograph equipped with a DB-1 column (100 m, 0.25 mm ID, 0.5μm film
thickness). Separated compounds were directed to a combustion interface (1/4" OD, 0.5 mm ID,
44 cm long ceramic tube with copper, nickel and platinum wires inside) where they were
converted to $H_2O$ and $CO_2$. $H_2O$ was removed by Nafion tubing, while $CO_2$ was directed into an
Isotope Ratio Mass Spectrometer (IRMS) where m/z 44, 45 and 46 were analysed. Performance
characteristics of the sampling and analysis procedures discussed by Kornilova et al. (2015a) are
summarized in Table 1. The accuracy of $\delta^{13}C$ measurements was 0.5 ‰ for sample masses
exceeding 3 ng of the target compound. For a typical 24h sample this mass corresponds to an
atmospheric mixing ratio of 20 to 30 pmol mol$^{-1}$. $\delta^{13}C$ results are not reported here for sample
masses that were lower than this.



Ambient VOC were collected at a site located in the Northern part of the Greater Toronto
Area (GTA) on the premises of the Downsview location of Environment Canada (43°46′N,
79°28′W) from 2009 to 2010 (Fig. 1). Samples were collected from Monday to Thursday over 24
h time periods (7 to 7 AM), Friday sampling lasted 6 to8 h (7 AM to 4 PM) and Friday to Sunday
samples were collected over 64 h (4 PM on Friday to 7 AM on Monday). Most samples were
collected between Fall 2009 and Winter 2010 (October 2009 to February 2010), with some
additional samples collected in March 2010, and from August 2010 to September 2010. The
average sampling flow rate was 24 to 25 mL min$^{-1}$. In total, 74 samples were analysed for VOC
concentrations and carbon isotope ratios, although in several of those samples carbon isotope
ratios could only be determined for some of the target compounds.
**3. Results and discussion**
**3.1 Overview**
Averages and some basic statistics for concentrations of ambient aromatic VOC collected
in Toronto (2009 to 2010) are summarized in Table 2. Mixing ratios determined for the targeted
VOC are in the low to mid pmol mol$^{-1}$ range and are similar to mixing ratios reported by
Canada's National Air Pollution Surveillance Program (NAPS, 2009-2010) for downtown
Toronto and two suburban locations in Toronto (Table 3). The observed mixing ratios are also
within the range of values reported in several measurement series at various locations in North
America (Kornilova et al., 2015b; Jobson et al., 2004; Pankow et al., 2003; Riemer et al., 1998;
Hagerman et al., 1997; Roberts et al., 1984).
Except for benzene, the VOC mixing ratios are linearly correlated with each other with y-
axis intercepts close to zero. Except for the linear correlation between the mixing ratios of
ethylbenzene and o-xylene, the y-axis intercept is statistically not different from zero (95%
confidence interval) (Table 4, Fig. 2). The VOC concentration ratios observed are within or close
to the range of values reported in literature. Exceptions are the toluene over benzene ratio and the
p,m-xylene over o-xylene ratio, which are higher than previously reported ratios (Table 5) but
similar to the ratios that can be derived from the NAPS data reported for other locations in
Toronto in Table 3.

7        Averages and basic statistics for measured carbon isotope ratios are summarized in Table

6.  The majority of the determined $\delta^{13}C$ values are comparable with those observed in other
ambient studies (Table 7). The only exception is the value reported by Redeker et al. (2007) for
benzene, which is about 3 ‰ lighter than the average for our measurements, and close to the 10[th]
percentile of our observations.

12        Figure 3 shows the frequency distribution for the measured carbon isotope ratios. The

lower end of the distribution is close to the carbon isotope ratios reported in literature for their
emissions (Table 8). The only exception are the benzene and toluene carbon isotope ratios
reported by Turner et al. (2006) for industrial stack emissions which are about 3 to 5 ‰ heavier
than the lower end of our observations. Turner et al. (2006) reported no details of the type of
industrial emissions studied and therefore the possible impact of such heavier industrial
emissions on ambient carbon isotope ratios cannot be evaluated. Only 13 of our 275 measured
carbon isotope ratios are significantly below the carbon isotope ratios reported by Rudolph et al.
(2002) for emissions in Toronto, whereas 186 observations are significantly heavier (95 %
confidence limit determined from the uncertainty of the emission ratios and measurement errors).

22        Overall, there is little systematic change of the carbon isotope ratios with season (Fig. 3).

The only significant exceptions are the carbon isotope ratios of the $C_2$-alkylbenzenes, which are



approximately 2 ‰ lighter in spring and fall compared to summer and winter (T-test, 99.9%
probability).

3         There is no clear dependence between mixing ratios and carbon isotope ratios (Fig. 5),

although in some cases the mixing ratios on average differed for different ranges of carbon
isotope ratios (see below). There is a strong dependence between carbon isotope ratios of the $C_2$-
alkylbenzenes and some weak correlations between the carbon isotope ratio of toluene and the
$C_2$-alkylbenzenes. No correlation between the carbon isotope ratios of benzene and any of the
other aromatic VOC is found (Fig. 4, Table 9). None of the y-axis intercepts of the linear
regressions are significantly different from zero (95 % confidence interval), which is consistent
with the finding that the carbon isotope ratios of emissions for different aromatic VOC are
similar (Rudolph et al., 2002).
**3.2 Ambient carbon isotope ratios and atmospheric VOC processing**

14        The observed carbon isotope ratios cover a range of more than 8 ‰ (Fig. 3), which

cannot be explained by measurement errors (Table 1) or uncertainty of the carbon isotope ratios
of known VOC sources (Table 8). Although the existence of unidentified sources of VOC with
very heavy carbon isotope ratios cannot be completely ruled out, it is unlikely that this could
explain the observed variability of measured carbon isotope ratios since such an unidentified
source would have to be of approximately the same magnitude as the known sources to explain
the absence of a systematic dependence between VOC carbon isotope ratio and VOC mixing
ratios (Fig. 5).

22        Anderson et al. (2004) reported that the reaction of aromatic VOC with the OH-radical,

which is the dominant loss process for light aromatic VOC in the atmosphere, results in

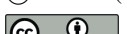



enrichment of $^{13}$C in the residual VOC. Varying extent of reaction with OH-radicals can
therefore explain the observed variability of carbon isotope ratios as well as the absence of a
significant number of $\delta^{13}$C values lower than the $\delta^{13}$C values of emissions (Table 8). Table 10
summarizes the rate constants and KIEs for the reaction of the studied compounds with the OH-
radical as well as the $\delta^{13}$C values reported for their emissions in Toronto by Rudolph et al.
(2002). Included in Table 10 are also the $\delta^{13}$C values calculated from $\delta^{13}C_{steady\ state} = \delta^{13}C_{emissions}$
$+\varepsilon_{OH}$ for steady state between emissions and loss by reaction with the OH-radical. For
comparison the average of measured $\delta^{13}$C values is also given in Table 11.

9        The averages of the measured $\delta^{13}$C values are consistently lighter than the predictions for

steady state. However, only for toluene and benzene, the least reactive of the light aromatic
VOC, this difference is statistically significant at the 99% confidence level. Since the
measurements were made at a location impacted by substantial emissions within a less than 10
km radius it is not surprising that for VOC with a short atmospheric residence time the average
carbon isotope ratio is closer to a steady state between emission and removal reaction than for
less reactive VOC. Indeed, the Pearson product moment correlation coefficient for the difference
between steady state and average measured isotope ratios and atmospheric residence time is
0.863 and the $R^2$ for a linear correlation is 0.745. However, this is only based on data for 5
different compounds and the average of the carbon isotope ratios is the result of isotope ratio
measurements covering a wide range of $\delta^{13}$C values (Fig. 3).

20       Table 11 shows a breakdown of measured $\delta^{13}$C values into different ranges by

compound. The number of observations with $\delta^{13}$C values heavier than steady state between
emission and removal is in the range of 25 % for the $C_2$-alkylbenzenes whereas for benzene and



toluene less than 10 % of the observations fall into this range. For benzene, the by far least
reactive of the studied VOC, the percentage of data points close to steady state $\delta^{13}C$ values is
even smaller. There is no systematic dependence between the percentages of $\delta^{13}C$ values close to
the source isotope ratios and VOC reactivity. This is compatible with the concept that
photochemical aging is an important process resulting in changes in the carbon isotope ratios of
ambient VOC as well as the predicted dependence between carbon isotope ratio and the rate
constant for the atmospheric VOC removal reactions (Equation 3).
In principle these findings are similar to the change in VOC concentration ratios as result
of photochemical processing, which has been used to study and quantify photochemical VOC
processing in the atmosphere (Kleinman et al., 2003; Jobson et al., 1999; Jobson et al., 1998;
Parrish et al., 1992; Rudolph and Johnen, 1990). It has been shown that the ∫[OH]dt determined
from $\delta^{13}C$ values of VOC is a valid approximation for the age of the studied VOC even in the
case of mixing air masses with different values for ∫[OH]$dt$ (Rudolph and Czuba, 2000), whereas
the use of VOC concentration ratios to quantify VOC processing requires an identical ∫[OH]$dt$
for VOC with different reactivity (Rudolph and Johnen, 1990), a condition that often is referred
to as photochemical age of the air mass (Parrish et al., 1992). The validity of the assumption of
an identical ∫[OH]$dt$ for VOC of different reactivity can be tested by a comparison of carbon
isotope ratios predicted by Equation 3 for identical ∫[OH]$dt$:

$$\frac{\delta_A^{A\,13}C - \delta_s^{A\,13}C}{\delta_A^{B\,13}C - \delta_s^{B\,13}C} = \frac{k_{OH}^A \varepsilon_{OH}^A}{k_{OH}^B \varepsilon_{OH}^B} \qquad \text{Eq. 4}$$

In Table 12 the slopes calculated for identical values for the average ∫[OH]$dt$ of different
VOC are compared with the slopes determined from linear regression of measured carbon





isotope ratios. With the exception of the dependence between toluene and ethylbenzene, which
have similar reactivity towards the OH-radical, the calculated slopes differ substantially from the
observed slopes demonstrating that the assumption of an identical $\int[OH]dt$ for VOC with
different reactivity is not valid.  This strongly points towards mixing air masses containing VOC
with different values for $\int[OH]dt$. This is consistent with the location of the sampling site at the
outskirts of a major city and a sampling duration that lasted at least several hours, in most cases
about one day.
**3.3 Photochemical age and mixing of air masses**

10          Carbon isotope ratios of VOC allow calculation of the average $\int[OH]dt$ of the individual

volatile organic compound without the requirement of a uniform $\int[OH]dt$ and therefore our
observations can be used to determine the $\int[OH]dt$ of VOC with different atmospheric life time
independent of assumptions about atmospheric mixing. Table 13 lists basic statistics for the
values of $\int[OH]dt$ determined from our $\delta^{13}C$ measurements based on Equation 3 using the kinetic
data and carbon isotope ratios of VOC emissions listed in Table 10.

16          Due to the uncertainty of the carbon isotope ratios of emissions and measurement errors

low values of $\int[OH]dt$ have large relative uncertainty. The consequence is a substantial number
of negative $\int[OH]dt$ values. Nevertheless, out of the 275 $\int[OH]dt$ values only 13 are below zero
at a 95 % confidence limit determined from the uncertainty of the $\int[OH]dt$ whereas 166 of the
$\int[OH]dt$ values are significantly larger than zero (95% confidence limit).

21          The values for average $\int[OH]dt$ of the different VOC differ substantially. The highest

average $\int[OH]dt$ of $3.1 \times 10^{11}$ s molecules $cm^{-3}$ is found for benzene, whereas the average $\int[OH]dt$
for the most reactive VOCs is only $3.4 \times 10^{10}$ s molecules $cm^{-3}$ (Table 13). There is strong



evidence for a systematic dependence between atmospheric residence time and average ∫[OH]*dt*
(Fig. 6). The Pearson product moment correlation coefficient for the dependence between
∫[OH]*dt* and atmospheric residence time is 0.981 and $R^2$ for a linear fit is 0.963. The correlation
is strongly influenced by the data point for the benzene. When excluding this data point the
Pearson product moment correlation coefficient is 0.760 and $R^2$ for a linear fit is 0.578. The
finding of differences in ∫[OH]*dt* for VOC with different reactivity is similar to the results of a
comparison of the ages of ethane and n-butane derived from carbon isotope ratios reported by
Saito et al. (2002) for the western North Pacific. They report that the age of ethane in most cases
is approximately an order of magnitude larger than the age of n-butane, which is similar to the
∫[OH]*dt* -reactivity dependence in Figure 6, although the range of VOC reactivity covered in this
paper does not include the low reactivity of ethane.

12          The dependence of the ∫[OH]*dt* on VOC reactivity is also seen for individual samples. In

Figure 7 the ∫[OH]*dt* of different VOC are compared with ∫[OH]*dt* determined for p,m-xylene,
the most reactive VOC. Since the ∫[OH]*dt* is a linear function of the carbon isotope ratios
(Equation 3) it is not surprising that the comparison showed strong correlations between the
values for ∫[OH]*dt* of the $C_2$-alkylbenzenes and no significant linear correlation between the
∫[OH]*dt* of benzene and ∫[OH]*dt* of p,m-xylene. With the exception of 4 data points for ∫[OH]*dt*
of toluene the values for ∫[OH]*dt* of less reactive VOC were consistently higher than those for
p,m-xylene, the most reactive of the studied VOC.

20          The systematic dependence of ∫[OH]*dt* on VOC reactivity can be explained by mixing of

air masses that have been subject to different extents of photochemical processing. During
photochemical aging VOC of high reactivity will be removed at a faster rate than VOC of low
reactivity. Consequently the relative contribution to VOC of low reactivity compared to VOC of



high reactivity increases with increased photochemical processing and mixing of air masses that
have been subject to different degrees of photochemical processing will result in average ∫[OH]$dt$
values that decrease with increasing VOC reactivity.
**3.4 Photochemical age and VOC concentration ratios**
For VOC with identical ∫[OH]$dt$ the change in concentration ratios can be predicted from
∫[OH]$dt$ (Kleinman et al., 2003; Parrish et al., 1992; Rudolph and Johnen, 1990) as long as their
emission ratios are similar. In case of mixing of air masses with different ∫[OH]$dt$, the change in
VOC concentration ratios depends on the relative contributions from the different air masses and
therefore is difficult to predict. The possibility to derive ∫[OH]$dt$ for VOC from their carbon
isotope ratios allows studying the dependence between VOC ratios and ∫[OH]$dt$ of the individual
compound. The only condition for this is a narrow range of carbon isotope ratios for the
emissions of the studied VOC.
Figure 8 shows some examples for the dependence between VOC concentration ratios
and ∫[OH]$dt$ of o-xylene and toluene. The ratios exhibit substantial scatter, especially for ratios
relative to benzene or toluene. Overall there is no clear dependence between ∫[OH]$dt$ and VOC
ratios. For many VOC pairs the ratios show substantial deviations from the dependence between
∫[OH]$dt$ and concentration ratios predicted for identical values for average ∫[OH]$dt$. However, for
VOC pairs with similar reactivity or low reactivity for both VOC prediction and observation
agree. This suggests that, similar to the dependence between VOC concentrations and carbon
isotope ratios (Fig. 5) mixing of air masses with different VOC concentration ratios plays a
major role in determining VOC ratios.





A change in VOC concentration ratios may be caused by varying impact from emission
sources with different emission ratios or different $\int[OH]dt$. With increasing photochemical age
the concentrations of a more reactive VOC will decrease faster than the concentration of a less
reactive VOC. In case of mixing air masses containing VOC with different $\int[OH]dt$ the relation
predicted by the hydrocarbon clock concept is no longer applicable, but qualitatively the ratios
for concentrations of more reactive VOC over those of less reactive VOC will still decrease with
increasing $\int[OH]dt$.

8        Figure 9 shows the relative change in the ratios for the concentration of different VOC

over the concentration of benzene averaged for 4 intervals of $\int[OH]dt$ derived from the p,m-
xylene carbon isotope ratio. Also shown is the relative change in $\int[OH]dt$ of benzene for the
same intervals. Although the averages have substantial uncertainties, there is evidence for a
systematic dependence. For the $C_2$ alkyl benzenes the difference between the lowest and highest
ratios is significant at the 90% confidence level (T-test). Furthermore, it is unlikely that the
observation of similar trends for all ratios is a coincidence. Most importantly, the observations
are incompatible with the expected change of VOC concentration ratios with increasing $\int[OH]dt$.
Benzene is the by far least reactive of these VOC; consequently photochemical processing is
expected to decrease the alkylbenzene over benzene concentration ratios, which is not observed.

18       Figure 10 shows the relative difference between the averages for all VOC concentration

ratios and the average concentration ratios for VOC without significant depletion of $^{13}C$ relative
to the isotope ratio of emissions. The separation between these two groups is based on the 95%
confidence limit calculated from measurement error and uncertainty of $\delta^{13}C$ of emissions (Table
10). For comparison the change in ratios predicted from the average $\int[OH]dt$ of the VOC in
Table 13 is included in Figure 10. For $C_2$-alkylbenzenes the predicted and observed changes in





concentration ratios agree within their uncertainties, although for the o-xylene over ethylbenzene
concentration ratio the high uncertainty does not allow a meaningful comparison. However,
VOC concentration ratios relative to benzene or toluene show a substantial difference between
prediction and observation. Moreover, for concentration ratios relative to benzene the samples
with negligible photochemical processing have ratios lower than the average, which can only be
explained by impact of sources with different VOC emission ratios. Specifically, it can be
concluded that sources directly impacting the sampling location on average have lower emission
ratios relative to benzene than emissions from sources located at a distance allowing for
photochemical aging of the VOC. This demonstrates that in this case the differences in emission
ratios are larger than the impact of photochemical aging.

11       Based on the location of the sampling site, traffic related emissions are expected to have a

major impact on VOC concentrations. Other potentially important sources are emissions from
industrial facilities. In Table 14 VOC emission ratios for transportation related emissions from
literature and industrial facilities in Ontario and the Toronto region reported by the National
Pollutant Release Inventory (NPRI) are compared with the average ratios for our measurements.
Although VOC concentration ratios in vehicular emissions cover a range of nearly a factor of 2,
the observed average ratios relative to benzene are consistently at the upper end or above this
range but close to emission ratios for industrial facilities in Ontario.  Since emission ratios
relative to benzene for industrial facilities in the Toronto area are much higher than the average
for Ontario (Table 14) differences in emission ratios from industrial facilities between Toronto
and the rest of Ontario can be ruled out as explanation for the lower concentration ratios relative
to benzene for samples with little photochemical processing. In contrast to the concentration
ratios relative to benzene, due to the similarity in emission ratios for industrial facilities and



traffic (Table 14) and the possible change of these ratios due to photochemical processing, it is
difficult to extract information about contributions from these two source types from
concentration ratios for alkyl benzenes without additional information.

4         Most of the major sources of light aromatic VOC in Ontario are located Southwest and

West of the sampling site (Fig. 11). Therefore insight into the impact of emissions from
industrial facilities on concentration ratios and carbon isotope ratios can be gained by filtering
the samples based on predominant wind direction during sampling. In Table 15 the average VOC
carbon isotope ratios for the two wind sectors are compared. With the exception of toluene the
average carbon isotope ratios for samples collected under the influence of airflow from North or
East is heavier than in samples with wind from the South or West sectors. For benzene this
difference is statistically significant at the 98% level (T-test) and for the C2-benzenes at a >90%
level. This is consistent with the existence of larger sources of aromatic VOC in the S+W sector
compared to the N+E sector (Fig. 11). The absence of a statistically significant difference in
isotope ratios for toluene is most likely due to the existence of a nearby major industrial source
of toluene east of the sampling site (Fig. 12).

16        Based on the difference in carbon isotope ratios between samples for the two wind

sectors it is expected that there also will be differences in concentration ratios. In Table 15
average VOC concentration ratios for samples dominated by southerly or westerly airflow are
compared with ratios for samples with winds from either North or East.  For ratios relative to
benzene substantial differences between the two wind sectors are found. For the $C_2$-
alkylbenzenes the difference is statistically significant at >95 % (T-test), for toluene at the 89 %
level. However, for concentration ratios between alkylbenzenes no statistically significant
differences can be seen. The lower concentration ratios relative to benzene in the N+E samples





can therefore be explained either by impact of aged air which is, relative to benzene, depleted in
the more reactive alkyl benzenes due to photochemical processing or lower emissions of
alkylbenzenes from industrial facilities in the N+E sector.

4        VOC carbon isotope ratios can be used to test which of these two factors dominates.

Using the carbon isotope ratio of p,m-xylene the N+E sector samples are separated into two
groups of samples with different values for ∫[OH]$dt$. The separation into two groups is based on
∫[OH]$dt$ of p,m-xylene being larger than zero at the 99% confidence limit. The confidence limit
was calculated from the uncertainty of the isotope ratio of emissions, the measurement
uncertainty and the uncertainty of the KIE. This ∫[OH]$dt$ is $3 \times 10^{10}$ OH-radicals cm$^{-3}$ s, which
corresponds to approximately 8 h for an average OH radical concentration of $10^6$ cm$^{-3}$. The
average concentration ratios for these two groups are included in Table 15. The average ∫[OH]$dt$
of p,m-xylene for samples dominated by recent emissions of p,m-xylene is $2.3 \times 10^9$ OH-radicals
cm$^{-3}$ s corresponding to 0.6 hours for an average OH radical concentration of $10^6$ cm$^{-3}$. However,
it should be noted that, due to the uncertainty in the isotope ratio of p,m-xylene emissions this
average value has an uncertainty of approximately $1.3 \times 10^{10}$ OH-radicals cm$^{-3}$ s corresponding to
nearly 4 hours for an average OH radical concentration of $10^6$ cm$^{-3}$. For samples dominated by
aged p,m-xylene the average ∫[OH]$dt$ is $7.4 \times 10^{10}$ OH-radicals cm$^{-3}$ s corresponding to 21 h for an
average OH radical concentration of $10^6$ cm$^{-3}$.

19       In contrast to the expected decrease in the concentration ratios for more reactive VOC

over less reactive VOC with increasing ∫[OH]$dt$, the average ratios are always higher for the aged
samples. Although the difference is statistically significant at a > 90 % level for only 6 of the
concentration ratios, this finding rules out photochemical processing as main reason for the
observed changes in VOC concentration ratios. For this set of observations differences in



∫[OH]*dt* derived from carbon isotope ratios most likely reflect differences in the footprint area
that impacts VOC concentration ratios. Samples with low values for ∫[OH]*dt* represent the
impact of nearby sources, most likely traffic related emissions, consistent with concentration
ratios that are close to the typical traffic dominated VOC ratios (Table 14).
The concentration ratio of p,m-xylene over ethylbenzene has been used as an indicator of
photochemical processing (Miller et al., 2012; Monod, 2001). It has been reported that the ratio
of p,m-xylene to ethylbenzene emissions is nearly constant for different sources (i.e. exhaust,
petrochemical activities, solvent use, etc.) and varies from 3.5 to 4.0 (Nelson and Quigley, 1983).
Since photochemical removal is approximately 3 times faster for p,m-xylene than for
ethylbenzene (Table 6), it has been suggested that any value of this ratio lower than 3.5 is a good
indicator of photochemical ageing (Nelson and Quigley, 1983; Miller et al., 2012; Monod, 2001;
Civan et al., 2015). This is in contrast to the higher concentration ratio of p,m-xylene to
ethylbenzene observed in samples with photo-chemically aged p,m-xylene for the N+E sector
compared to samples dominated by recent emission of p,m-xylene (Table 15). Although the
observed relative increase in concentration ratios of 50% has an uncertainty of 40% and is
statistically not significant (T-test), it is inconsistent with the decrease by approximately 60%
which would is expected from the difference in the average ∫[OH]*dt* for these two data sets. This
can be explained by the difference in xylene over ethylbenzene emission ratios between traffic
related emissions and industrial facility emissions (Table 14). Samples with low values for the
average ∫[OH]*dt* for p,m-xylene will be dominated by emissions from nearby sources, most
likely traffic, while for samples with higher ∫[OH]*dt* for p,m-xylene the footprint area will be
much larger and industrial emissions will play a more significant role.





**4. Conclusions**
Measurement of carbon isotope ratios of ambient VOC can be used to obtain detailed
insight into origin and sources of VOC. This is especially important in cases where different
source types impact ambient VOC levels and the source location alone does not allow identifying
the dominant source type. In such cases the use of VOC concentration ratios can result in
incorrect estimates of photochemical processing. Moreover, the comparison of $\int[OH]dt$ derived
from carbon isotope ratios of VOC demonstrates that in the case of mixing of air masses with
differences in $\int[OH]dt$ the average $\int[OH]dt$ for VOC with different reactivity differ. Typically
$\int[OH]dt$ for more reactive VOC will be lower than $\int[OH]dt$ for less reactive VOC. It can
therefore be concluded that in this study the assumption of a uniform $\int[OH]dt$ for different VOC
is not justified and consequently the use of VOC concentration ratios to determine $\int[OH]dt$
would give incorrect results.
The use of carbon isotope ratios or $\int[OH]dt$ derived from carbon isotope ratios allows
differentiation between samples impacted by emissions from different footprint areas.  The
comparison of carbon isotope ratios and VOC concentration ratios demonstrates that in this study
differences in the emission ratios for different types of VOC sources have a larger impact on
ambient VOC concentration ratios than photochemical aging.
For the VOC studied here reaction with the OH-radical is the only relevant loss process
in the atmosphere. Although the OH-radical concentration exhibits significant seasonal, diurnal
and spatial variability, $\int[OH]dt$ derived from isotope ratios also depend on the time between
emission of the VOC and observation. Consequently these $\int[OH]dt$ also contain information on
the footprint area which impacts the observation sites. VOC have widely different photochemical
reactivity; the reactivity of VOC used in our study cover a range of a factor of nearly 20. Values



for average ∫[OH]$dt$ derived from carbon isotope ratios cover the range from effectively zero to
approximately $8 \times 10^{11}$ OH-radicals cm$^{-3}$ s corresponding to 9 days for an average OH radical
concentration of $10^6$ cm$^{-3}$. The lower end of useful values for ∫[OH]$dt$ that can be determined
from VOC carbon isotope ratios strongly depends on the uncertainty of the carbon isotope ratio
of emissions. Due to the limited number of studies of carbon isotope ratios of VOC emissions,
this substantially contributes to uncertainty in isotope ratio derived ∫[OH]$dt$. For the most
reactive VOC used in this study the uncertainty in ∫[OH]$dt$ resulting from uncertainty in the
carbon isotope ratio corresponds to 4 h at an average OH radical concentration of $10^6$ cm$^{-3}$. The
uncertainty in ∫[OH]$dt$ of p,m-xylene due to carbon isotope ratio measurement errors corresponds
to less than 1.5 h. Therefore more detailed understanding of the carbon isotope ratio of VOC
emissions will reduce the lower end of meaningful values for ∫[OH]$dt$ that can be determined
from carbon isotope ratios of atmospheric VOC.

13         In this study sampling periods of at least several hours, in most cases of one day were

used. While this is useful for determining meaningful averages from a limited set of
measurements, this does not allow differentiating between changes during sampling and mixing
of air masses. The finding of a dependence between average ∫[OH]$dt$ and VOC reactivity is
similar to the finding of a substantial difference in ∫[OH]$dt$ between ethane and n-butane in a
marine environment (Saito et al., 2002) as well as global modelling studies (Thompson et al.,
2003; Stein and Rudolph, 2007).  However, to our knowledge, apart from this study no data that
would allow systematic study of the relation between VOC concentration ratios and carbon
isotope ratios in urban areas have been published. It can therefore not be decided to which extent
the observed discrepancy between ∫[OH]$dt$ derived from carbon isotope ratios and changes in





VOC concentration ratios is a general problem for determining $\int[OH]dt$ from VOC concentration
ratios or limited to conditions specific for this or similar data sets.
**Acknowledgements**
The authors would like to thank Darrell Ernst and Wendy Zhang at Environment Canada for
technical support in isotope ratio measurements. This research was supported financially by the
Natural Sciences and Engineering Research Council of Canada (NSERC) and the Canadian
Foundation for Climate and Atmospheric Sciences (CFCAS).

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





**Table 1.** Performance of the system for analysis of volatile organic compounds in ambient air.

| Compound | Detection Limit[a] (ng) | Relative precision in concentration measurement[b] (%) | Precision in $\delta^{13}C$ measurement[c] (‰) |
|---|---|---|---|
| Benzene | 1.5 | 4 | 0.3 |
| Toluene | 0.4 | 4 | 0.2 |
| Ethylbenzene | 0.6 | 6 | 0.2 |
| o-Xylene | 0.4 | 8 | 0.3 |
| p,m-Xylene | 0.5 | 9 | 0.3 |

[a] Detection limit calculated using $3\sigma$, where $\sigma$ is the standard deviation of blank values
determined from 3-4 repeat measurements.
[b] Precision calculated as relative standard deviation (%) of peak area for >10 repeat
measurements.
[c] Precision calculated as standard deviation of $\delta^{13}C$ values for >10 repeat measurements.
**Table 2.** Mixing ratios (nmol mol$^{-1}$) determined for volatile organic compounds collected in
Toronto (2009-2010).

| Compound | N[a] | Mean | s[b] | Median | Minimum | Maximum | 90th percentile | 10th percentile |
|---|---|---|---|---|---|---|---|---|
| Benzene | 65 | 0.13 | 0.12 | 0.11 | 0.02 | 0.74 | 0.28 | 0.03 |
| Toluene | 73 | 0.64 | 0.44 | 0.56 | 0.05 | 2.11 | 1.39 | 0.16 |
| Ethylbenzene | 69 | 0.07 | 0.05 | 0.05 | 0.01 | 0.26 | 0.14 | 0.02 |
| o-Xylene | 66 | 0.05 | 0.03 | 0.04 | 0.01 | 0.17 | 0.10 | 0.01 |
| p,m-Xylene | 68 | 0.17 | 0.13 | 0.14 | 0.01 | 0.74 | 0.36 | 0.04 |

[a] Number of measurements.
[b] Sample standard deviation.
**Table 3.** Average mixing ratios (nmol mol$^{-1}$) calculated from data reported by the National Air
Pollution Surveillance Program (NAPS) for three locations in Toronto during the time periods
covered in this work.

| Location | Etobicoke West[a] | | Downtown Toronto[b] | | Etobicoke South[c] | | This work | |
|---|---|---|---|---|---|---|---|---|
| Compound | mean | s[d] | mean | s[d] | mean | s[d] | mean | s[d] |
| Benzene | 0.18 | 0.07 | 0.23 | 0.10 | 0.20 | 0.10 | 0.13 | 0.12 |
| Toluene | 0.49 | 0.31 | 0.62 | 0.55 | 0.68 | 0.65 | 0.63 | 0.44 |
| Ethylbenzene | 0.06 | 0.04 | 0.07 | 0.04 | 0.09 | 0.06 | 0.06 | 0.17 |
| o-Xylene | 0.05 | 0.04 | 0.07 | 0.04 | 0.07 | 0.05 | 0.05 | 0.03 |
| p,m-Xylene | 0.18 | 0.13 | 0.23 | 0.13 | 0.27 | 0.21 | 0.17 | 0.13 |

[a] Elmcrest Road, Toronto, ON, Canada.
[b] 223 College S., Toronto, ON, Canada.
[c] 461 Kipling Ave., Toronto, ON, Canada.
[d] Sample standard deviation.


**Table 4.** Linear regression analysis of mixing ratio correlations. Values above the diagonal are
$R^2$ and below are the intercepts[a].

|  | Benzene[b] | Toluene | Ethylbenzene | o-Xylene | p,m-Xylene |
|---|---|---|---|---|---|
| Benzene[b] |  | <0.3 | <0.3 | <0.3 | <0.3 |
| Toluene | NA |  | 0.495 | 0.586 | 0.488 |
| Ethylbenzene | NA | 0.011 (0.008) |  | 0.825 | 0.964 |
| o-Xylene | NA | 0.003 (0.006) | 0.008 (0.003) |  | 0.877 |
| p,m-Xylene | NA | 0.020 (0.023) | -0.007 (0.005) | -0.014 (0.011) |  |

[a] Standard error of intercept (nmol mol$^{-1}$) in parenthesis. The values are for linear fits of the
mixing ratio of the more reactive VOC versus the mixing ratio of the less reactive VOC.
[b] No significant correlation was found for linear regressions of the mixing ratios of any of the
VOC versus the mixing ratios of benzene.
**Table 5.** Ratios of VOC mixing ratios. Values below the diagonal are the ratios for the
measurements presented in this work[a], values above the diagonal give the range of ratios
observed in literature[b].

|  | Benzene | Toluene | Ethylbenzene | o-Xylene | p,m-Xylene |
|---|---|---|---|---|---|
| Benzene |  | 1.5-3.7 | 0.11-1.2 | 0.23-3.2 | 0.46-0.3.8 |
| Toluene | 6.6 (0.6) |  | 0.07-0.35 | 0.14-0.86 | 0.27-1.1 |
| Ethylbenzene | 0.69 (0.08) | 0.080 (0.01) |  | 0.6-2.1 | 1.2-4.3 |
| o-Xylene | 0.52 (0.07) | 0.069 (0.007) | 0.64 (0.04) |  | 1.7-2.9 |
| p,m-Xylene | 1.8 (0.2) | 0.22 (0.003) | 2.7 (0.07) | 3.7 (0.2) |  |

[a] Standard error of ratios in parenthesis. The ratios are for the mixing ratio of the more reactive
VOC over the mixing ratio of the less reactive VOC. For compounds with correlated mixing
ratios the slope of the linear regressions and standard errors of the slopes is given, in the absence
of correlation the average of the ratios and standard error of average ratios is given.
[b] Range of observations in urban, residential and commercial areas reported by Hoque et al.,
(2008), Karl et al. (2009), Miller et al. (2011), Alghamdi et al. (2014), de Gennaro et al. (2015)
and Rad et al. (2014).
**Table 6.** Stable carbon isotope composition ($\delta^{13}$C, ‰) of ambient VOC collected in Toronto
20  (2009-2010).

| Compound | N[a] | Mean | s[b] | Median | Range | 10th percentile | 90th percentile |
|---|---|---|---|---|---|---|---|
| Benzene | 44 | -25.0 | 3.4 | -26.0 | -39.4 to -13.6 | -28.4 | -20.9 |
| Toluene | 73 | -24.8 | 3.2 | -25.7 | -28.5 to -7.7 | -27.4 | -20.5 |
| Ethylbenzene | 58 | -24.0 | 4.0 | -23.4 | -34.8 to -17.4 | -28.5 | -18.8 |
| o-Xylene | 44 | -23.3 | 3.5 | -23.3 | -30 to -16.5 | -28.0 | -18.4 |
| p,m-Xylene | 56 | -24.0 | 4.1 | -23.8 | -34.6 to -16.4 | -28.4 | -18.9 |

[a] Number of measurements.
[b] Sample standard deviation.



**Table 7.** Average and standard deviation of reported stable carbon isotope composition measurements ($\delta^{13}$C, ‰) of ambient light
aromatic VOC[a].

| Location | Benzene | Toluene | Ethylbenzene | o-Xylene | p,m-Xylene |
|---|---|---|---|---|---|
| Toronto[b] | -24.6 (2.3) | -25.0 (1.1) | -25.3 (2.7) | -24.9 (1.5) | -25.6 (1.0) |
| Belfast[c] | -23.8 (2.5) | -26.9 (0.9) | -26.5 (1.4) | -26.3 (1.3) | -27.4 (1.7) |
| Crossgar and Hillsborough[d] | -28.3 (1.7) | | | | |
| South-west Germany[e] | | -27 to -23 | | | |
| Egbert[f] | -25.3 (2.6) | -24.8 (1.7) | -23.7 (3.6) | -23.4 (2.6) | -23.8 (3.0) |
| Yurihonjo[g] (Japan) | | | -29.6 to -23.5 | | |

[a] Sample standard deviation is given in parenthesis.
[b] Urban Canada (Rudolph et al., 2002).
[c] Urban Northern Ireland (Redeker et al., 2007).
[d] Rural Northern Ireland (Redeker et al., 2007).
[e] Vertical profile over rural area, only range of data available (Wintel et al, 2013).
[f] Rural Canada (Kornilova et al., 2015b).
g Urban Japan, only range of data available (Kawashima and Murakami, 2014).





**Table 8.** Average stable carbon isotope ratios ($\delta^{13}$C, ‰) of major sources of ambient VOC.
The sample standard deviation is given in parenthesis.

| Source | Benzene | Toluene | Ethylbenzene | o-Xylene | p,m-Xylene |
|---|---|---|---|---|---|
| Gasoline[a] | -27.0 (-24.5 to-29.9) [b] | -26.0,-28.1[b] | -27.4 (2.8) | -26.5 (1.9) | -27.9 (1.2) |
| Tunnel[c] | -26.5 (1.0) | -27.5 (1.0) | -27.4 (0.9) | -27.3 (0.4) | -26.9 (2.0) |
| Gas Station[c] | -29.1 (0.3) | -27.4 (0.6) | -28.2 (0.4) | -27.1 (0.6) | -27.7 (0.5) |
| Gas Station[d] | -27.35 (1.6) | -27.08 (0.7) | -26.48 (1.6) | | -27.4 (0.8) |
| Underground garage[c] | -27.7 (0.7) | -27.1 (0.7) | -27.5 (1.1) | -27.2 (1.1) | -27.7 (1.0) |
| Biomass burning[e] | -26.0 (0.1) | -26.5 (0.9) | -25.7 (0.5) | | |
| Biomass burning[f] | -27.6 (1.6) | -27.1 (1.3) | | | |
| Fossil fuel combustion[f] | -26.9 (0.3) | -27.5 (0.6) | | | |
| Industrial stack[g] | -23.5 (0.11) | -25.4 (0.48) | | | |

[a] Averages and standard deviations were calculated from the results of analysis of 4 individual
gasoline samples reported by Smallwood et al. (2002).
[b] For benzene only average and range are available, for toluene only two individual values were
reported.
[c] (Rudolph et al., 2002).
[d] (Kawashima and Murakami, 2014).
[e] (Czapiewski et al., 2003).
[f] (Giebel et al., 2010).
[g] (Turner et al., 2006).
**Table 9.** Regression analysis of delta values. Values above diagonal are $R^2$, and below are the
intercepts.

| | Benzene[b] | Toluene | Ethylbenzene | o-Xylene | p,m-Xylene |
|---|---|---|---|---|---|
| Benzene[b] | | <0.3 | <0.3 | <0.3 | <0.3 |
| Toluene | NA | | 0.361 | 0.356 | 0.335 |
| Ethylbenzene | NA | -0.04 (4.31)[a] | | 0.820 | 0.963 |
| o-Xylene | NA | -2.58 (4.31) | -2.54 (1.56) | | 0.825 |
| p,m-Xylene | NA | -0.49 (4.57) | -0.19 (0.64) | -1.64 (1.63) | |

[a] Standard error of intercept (‰) in parenthesis. The values are for linear fits of the mixing ratio
of the more reactive VOC versus the mixing ratio of the less reactive VOC.
[b] No significant correlation was found for linear regressions of the mixing ratios of any of the
VOC versus the mixing ratios of benzene.



**Table 10**. Summary of kinetic data, carbon isotope ratio of emissions, carbon isotope ratio
calculated for steady state between emission and loss, and average measured carbon isotope ratio
for light aromatic VOC[a].

| Compound | $k_{OH}$[b] ($10^{-12}$ cm$^3$ molec$^{-1}$ s$^{-1}$) | $\varepsilon_{OH}$ (‰)[c] | $\delta^{13}$C source (‰)[d] | $\delta^{13}$C steady state (‰) | Average of $\delta^{13}$C measured (‰) |
|---|---|---|---|---|---|
| Benzene | 1.22 | 7.83 (0.42)[e] | -28.0 (0.2) | -20.1 (0.5) | -25.0 (0.5) |
| Toluene | 5.63 | 5.59 (0.28) | -27.6 (0.5) | -21.7 (0.5) | -24.8 (0.4) |
| Ethylbenzene | 7.0 | 4.34 (0.28) | -27.7 (0.2) | -23.4 (0.4) | -24.0 (0.5) |
| o-Xylene | 13.6 | 4.27 (0.05) | -27.2 (0.2) | -22.9 (0.2) | -23.3 (0.5) |
| p,m-Xylene | 20.5[f] | 4.83 (0.05)[g] | -27.4 (0.4) | -22.6 (0.4) | -24.0 (0.6) |

[a]Values in parenthesis are the standard error of the mean.
[b] Rate constants for 298 K (Atkinson and Arey, 2003;Finlayson-Pitts and Pitts, 2000).
[c] Carbon kinetic isotope effects for reaction with OH radicals (Anderson et al., 2004a;Anderson
et al., 2004b;Rudolph et al., 2002).
[d] Isotopic composition of traffic related VOC emissions in Toronto  (Rudolph et al., 2002).
[e] Average ε calculated from 8.13 (0.8) (Anderson et al., 2004b) and 7.53 (0.5) (Rudolph et al.,
10  2002).
[f] Average of $k_{OHp-xylene}=1.43\times10^{-11}$ cm$^3$ molec$^{-1}$ s$^{-1}$ and $k_{OHm-xylene}=2.36\times10^{-11}$ cm$^3$ molec$^{-1}$ s$^{-1}$.
[g] $\varepsilon_{OH}$ (‰) is for p-xylene, no value is available for m-xylene.
**Table 11.** Overview of the number of observations within different ranges of carbon isotope
ratios[a].

| Range for $\delta^{13}$C | Benzene | Toluene | Ethylbenzene | o-Xylene | p,m-Xylene |
|---|---|---|---|---|---|
| $<\delta^{13}C_{source}$ [b] | 1 (2%) | 0 | 4 (7%) | 4 (9%) | 4 (7%) |
| $\approx \delta^{13}C_{source}$ [c] | 10 (23%) | 32 (44%) | 14 (24%) | 5 (11%) | 15 (27%) |
| $> \delta^{13}C_{source} < \delta^{13}C_{global}$ [d] | 28 (64%) | 15 (21%) | 4 (7%) | 6 (14%) | 4 (7%) |
| $\approx\delta^{13}C_{steady\ state}$ [e] | 2 (4 %) | 23 (31%) | 19 (33%) | 17 (39%) | 21 (38%) |
| $> \delta^{13}C_{steady\ state}$ [f] | 3 (7%) | 3 (4 %) | 17 (29%) | 12 (27%) | 12 (21%) |
| Total | 44 | 73 | 58 | 44 | 56 |

[a] The value in parenthesis gives the percentage of observation. The confidence intervals used to
determine the $\delta^{13}$C ranges were calculated from measurement error, uncertainty of $\delta^{13}$C of
emissions, and uncertainty of $\varepsilon_{OH}$.
[b] Number of $\delta^{13}$C measurements below the 95% confidence limit of $\delta^{13}$C of emissions.
[c] Number of $\delta^{13}$C measurements within the 95% confidence limit of $\delta^{13}$C of emissions.
[d] Number of $\delta^{13}$C measurements above the 95% confidence limit of $\delta^{13}$C of emissions but below
the 95% confidence limit of $\delta^{13}$C calculated for steady state between emissions and loss.
[e] Number of $\delta^{13}$C measurements within the 95% confidence limits of $\delta^{13}$C calculated for steady
state between emissions and loss.
[f] Number of $\delta^{13}$C measurements above the 95% confidence limit of $\delta^{13}$C calculated for steady
state between emissions and loss.



**Table 12.** Comparison of slopes calculated from Eq. 4 based on the assumption that different
VOC have identical ∫[OH]$dt$ (above diagonal) with slopes determined from linear regression of
the observations (below diagonal).

|  | Benzene | Toluene | Ethylbenzene | o-Xylene | p,m-Xylene |
|---|---|---|---|---|---|
| Benzene |  | 3.29 | 3.18 | 6.07 | 6.08 |
| Toluene | NA[a] |  | 0.97 | 3.15 | 1.85 |
| Ethylbenzene | NA | 0.96 (0.17)[b] |  | 3.26 | 1.91 |
| o-Xylene | NA | 0.85 (0.18)[b] | 0.89 (0.07)[b] |  | 0.59 |
| p,m-Xylene | NA | 0.95 (0.18)[b] | 1.0 (0.03)[b] | 0.93 (0.07)[b] |  |

[a] Not available, no correlation found.
[b] Standard error in parenthesis.
**Table 13.** Overview of ∫[OH]$dt$ [a] of light aromatic VOC calculated from their stable carbon
isotope ratios.

| Compound | Mean[b] | Median[c] | 90th percentile | 75th percentile | 25th percentile | 10th percentile |
|---|---|---|---|---|---|---|
| Benzene | 31 (5) | 21(14) | 76 | 49 | 11 | -4.1 |
| Toluene | 8.5 (1.1) | 5.7 (8) | 21 | 12 | 1.6 | 0.61 |
| Ethylbenzene | 12.1 (1.7) | 14 (5.7) | 29 | 20 | 2.3 | -2.8 |
| o-Xylene | 6.8 (0.9) | 6.8 (2.6) | 15 | 10 | 1.9 | -1.5 |
| p,m-Xylene | 3.4 (0.6) | 3.6 (2.4) | 8.7 | 6.5 | 0.5 | -1.0 |

[a] ∫[OH]$dt$ in $10^{10}$ s molecules cm$^{-3}$.
[b] Standard error of mean in parenthesis.
[c] 95% confidence interval in parenthesis.





**Table 14.** Comparison of observed VOC concentration ratios (mol/mol) with VOC emission
ratios for industrial facilities and mobile transportation.

| VOC concentration ratios | Facilities Ontario[a] | Facilities Toronto[a] | Gasoline engines[b] | Traffic Sarnia[c] | Traffic 8 European cities[d] | Average all observations |
|---|---|---|---|---|---|---|
| Toluene/Benzene | 6.2 | 120 | 2.6±1.9 | 3.6 | 2.0±0.4 | 6.6±0.56 |
| Ethylbenzene/Benzene | 0.68 | 13 | 0.20±0.19 | 0.57 | 0.38±0.08 | 0.69±0.083 |
| ∑ Xylenes/Benzene | 5.3 | 114 | 0.65±0.50 | 2.3 | 1.6±0.4 | 2.3±0.17 |
| ∑ Xylenes/Toluene | 0.86 | 0.96 | 0.34±0.28 | 0.63 | 0.81±0.2 | 0.35±0.019 |
| ∑ Xylenes/Ethylenzene | 7.9 | 8.6 | 4.0±1.3 | 4.1 | 4.1±0.5 | 3.5±0.096 |

[a] Calculated from emissions reported by the National Pollution Release Inventory (NPRI) for
2009 and 2010.
[b] Average and standard deviation calculated from 16 measurements of the composition of
exhaust from of gasoline engines using ethanol free or low ethanol fuel reported by the United
States Environmental Protection Agency, Speciate Data Base.
[c] Calculated from average ratios reported by Miller et al. (2011) for ambient observations
dominated by traffic related emissions in Sarnia (Southern Ontario, Canada).
[d] Calculated from mixing ratios reported by Monod et al. (2001) for observations dominated by
traffic related emissions in 8 major cities in Europe.



**Table 15.** Comparison of observed VOC concentration ratios (mol/mol) for different dominant
wind directions and different $\int[OH]dt$.

| VOC concentration ratios | S+W[a] | N+E[b] | N+E Recent[c] | N+E Aged[c] |
|---|---|---|---|---|
| Toluene/Benzene | 8.2±1.5 | 6.1±0.90 | 4.9±2.2 | 7.7±1.6 |
| Ethylbenzene/Benzene | 1.0±0.2 | 0.61±0.10 | 0.37±0.11 | 0.82±0.16 |
| o-Xylene/Benzene | 0.76±0.17 | 0.44±0.09 | 0.24±0.06 | 0.68±0.15 |
| p,m-Xylene/Benzene | 2.8±0.67 | 1.5±0.27 | 0.81±0.22 | 2.2±0.44 |
| Ethylbenzene/Toluene | 0.11±0.01 | 0.13±0.04 | 0.074±0.012 | 0.17±0.074 |
| o-Xylene/Toluene | 0.084±0.012 | 0.069±0.008 | 0.048±0.012 | 0.084±0.014 |
| p,m-Xylene/Toluene | 0.29±0.039 | 0.029±0.078 | 0.16±0.03 | 0.42±0.16 |
| o-Xylene/Ethylbenzene | 0.93±0.24 | 0.94±0.23 | 0.66±0.07 | 1.2±0.51 |
| p,m-Xylene/Ethylbenzene | 2.5±0.14 | 2.7±0.37 | 2.2±0.15 | 3.3±0.80 |
| p,m-Xylene/o-Xylene | 3.6±0.27 | 3.3±0.23 | 3.4±0.20 | 3.1±0.28 |
| $\sum$ Xylenes/Benzene | 3.3±0.81 | 1.8±0.34 | 1.0±0.29 | 2.7±0.55 |
| $\sum$ Xylenes/Toluene | 0.35±0.051 | 0.34±0.076 | 0.21±0.042 | 0.49±0.15 |
| $\sum$ Xylenes/Ethylenzene | 3.3±0.17 | 3.5±0.34 | 2.8±0.21 | 4.1±1.1 |

[a] Average ratio for data with winds dominantly from southern or western sectors. Only data
points for which at least during 80% of the sampling time the wind direction was within ± 45° of
the dominant wind direction are included. The number of data points is between 14 and 20. The
uncertainty stated is the standard error of the mean.
[b] Average ratio for data with winds dominantly from northern or eastern sectors are included.
Only data points for which at least during 80% of the sampling time the wind direction was
within ± 45° of the dominant wind direction are included. The number of data points is between
15 and 20. The uncertainty stated is the standard error of the mean.
[c] Data for the N+E sector filtered based on $\int[OH]dt$ determined from the p,m- xylene carbon
isotope ratio. Aged refers to samples with $\int[OH]dt > 0$ at the 99% confidence limit. The
confidence limit was calculated from the uncertainty of the isotope ratio of emissions, the
measurement uncertainty and the uncertainty of the KIE. Recent represents samples with $\int[OH]dt$
below this limit. The values given are the average ratios and the standard error of the mean. The
number of data points for Recent is 5, for Aged there are between 7 and 9 values.



**Figure Captions**
**Figure 1**. Location of sampling site in Toronto (S). Also shown are the locations for VOC
sampling by the National Air Pollution Surveillance Program (NAPS) in Toronto (A: Elmcrest
Road, Toronto, ON, Canada, B: 223 College St., Toronto, ON, Canada, C: 461 Kipling Ave.,
Toronto, ON, Canada. The map was produced using ©Google Earth, 2015.
**Figure 2.** Frequency distribution for VOC carbon isotope ratios measured in Toronto in 2009
and 2010. Observations for different seasons are identified as follows: Fall- dotted, winter- black,
and spring and summer- hatched. The solid vertical lines are the isotope composition of traffic
related emissions (Rudolph et al., 2002).
**Figure 3**. Example plots for the dependence between mixing ratios for different VOC pairs.
**Figure 4**. Example plots for the dependence between carbon isotope ratios for different VOC
pairs.
**Figure 5**.Example plots of VOC mixing ratios versus carbon isotope ratios.
**Figure 6**. Dependence between average $\int[OH]dt$ and photochemical life time for light aromatic
VOC. The average tropospheric life time was calculated using the rate constants given in Table
10 and an average tropospheric concentration of $1.16\times10^6$ OH-radicals cm$^{-3}$ (Spivakovsky et al.,
2000). Vertical error bars are the standard error of the mean.
**Figure 7.** Dependence between the $\int[OH]dt$ of different VOC and the $\int[OH]dt$ of p,m-xylene.
Only data with $\int[OH]dt$ larger than zero at a 95 % confidence limit are shown.
**Figure 8.** Plot of VOC concentration ratios versus $\int[OH]dt$ determined from carbon isotope
ratios. The lines show the change in carbon isotope ratios predicted for identical $\int[OH]dt$ of the
different VOC.
**Figure 9.** Relative change of VOC concentration ratios and $\int[OH]dt$ of benzene for different
ranges of $\int[OH]dt$ (in units of $10^{10}$ s molecules cm$^{-3}$) derived from the carbon isotope ratio of
p,m-xylene. The bars represent the mean concentration ratios normalized relative to the average
of all available ratios. The error bars are the relative standard errors of the mean. The range of
$\int[OH]dt$ for the lowest $\int[OH]dt$ bin was chosen to include all $\int[OH]dt$ which are not significantly
larger than 0 (95 % confidence limit). The other ranges were chosen to include between 8 and 12
data points in each bin.
**Figure 10.** Relative difference between the average of all VOC concentration ratios (mol/mol)
and the average VOC concentration ratios for VOC with carbon isotope ratios that are not higher
than the carbon isotope ratios of emissions (95 % confidence limit) (squares).  The circles show
the relative change expected for a uniform $\int[OH]dt$ calculated from the average $\int[OH]dt$ of all



observations (Table 13). The error bars show the standard error calculated from error
propagation. For data points with error bars smaller that the size of the data point no error bars
are show.
**Figure 11.** Location of facilities in southern Ontario with significant emissions of xylenes. The
numbers represent the annual emissions in tons year$^{-1}$. Emission data were taken from the
National Pollutant Release Inventory (NPRI). Only sources contributing at least 5% to the total
emissions are shown. Also shown is the location of the sampling site. The map was produced
using ©Google Earth, 2015
**Figure 12**. Location of facilities in Toronto with significant emissions of toluene. The numbers
represent the annual emissions in ton year$^{-1}$. Emission data were taken from the National
Pollutant Release Inventory (NPRI). Only sources contributing at least 5 % to the total emissions
are shown. Also shown is the location of the sampling site. The map was produced using
©Google Earth, 2015.



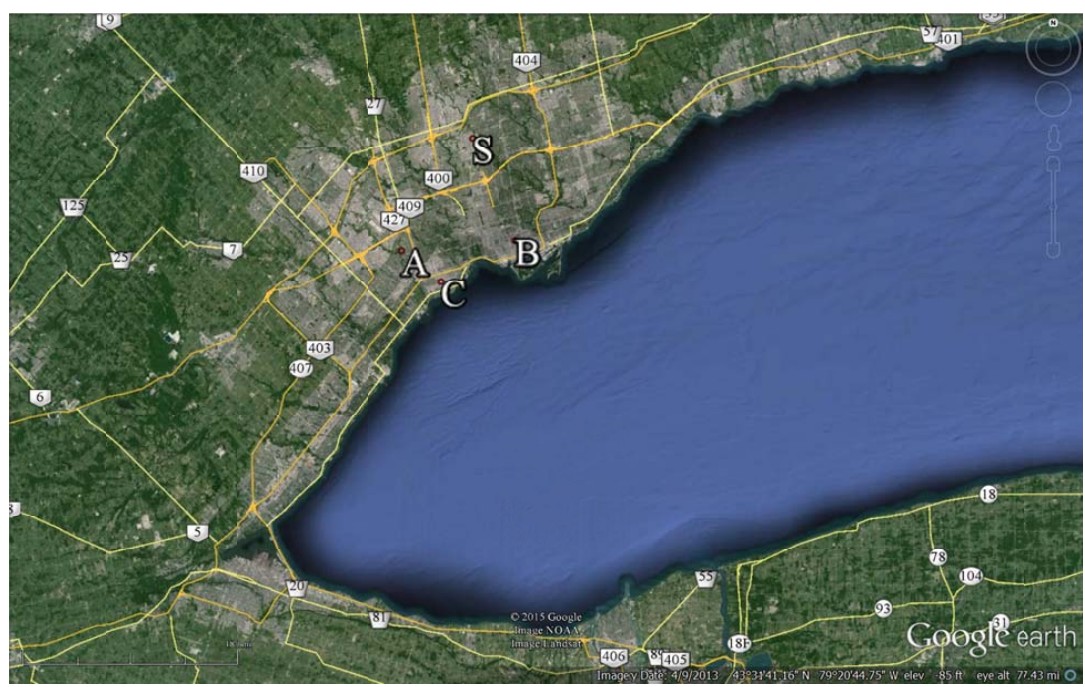

5      Figure 1



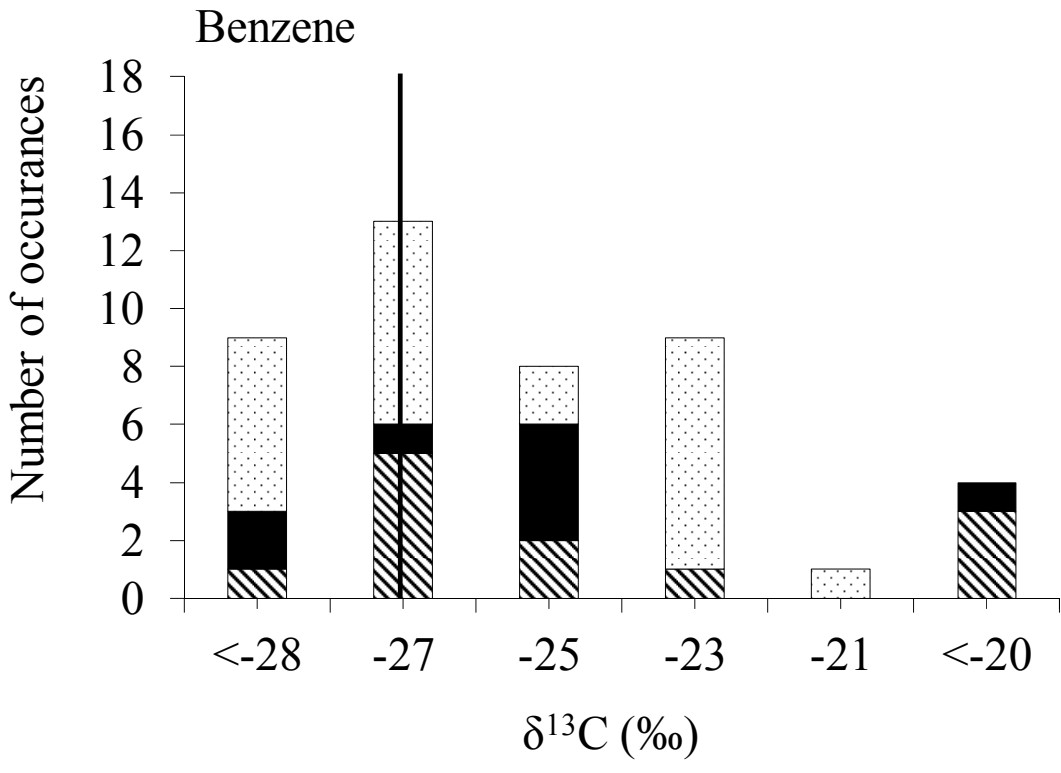

2   Figure 2





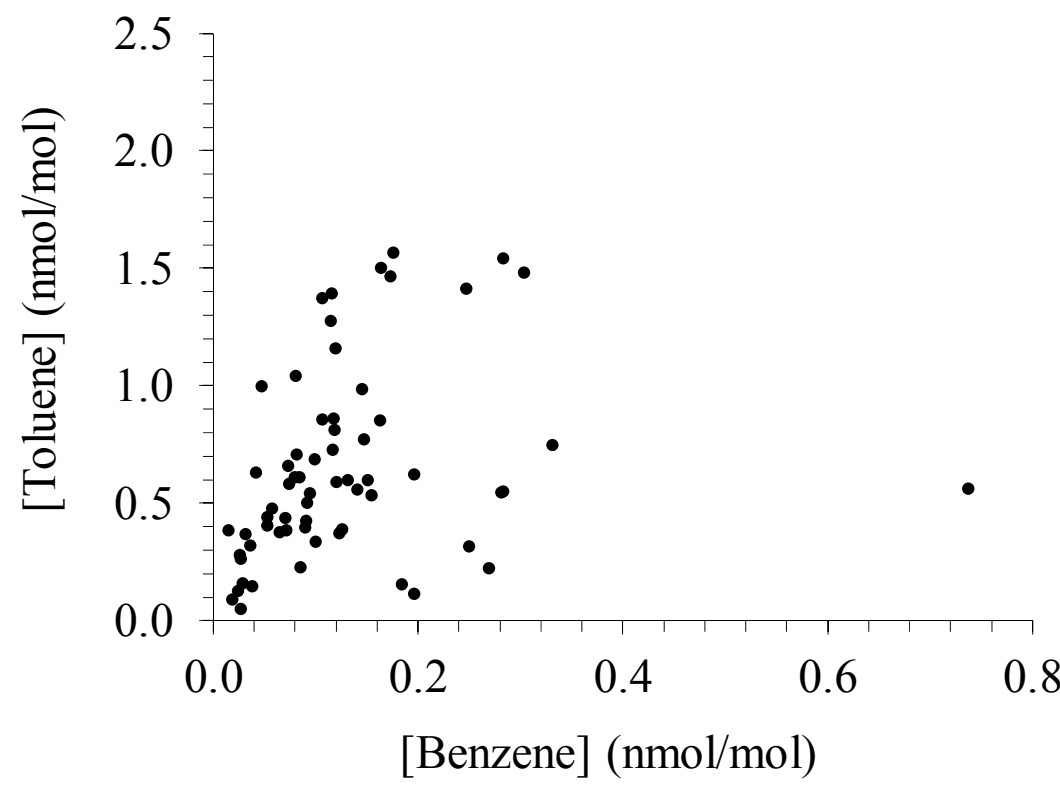

2    Figure 3




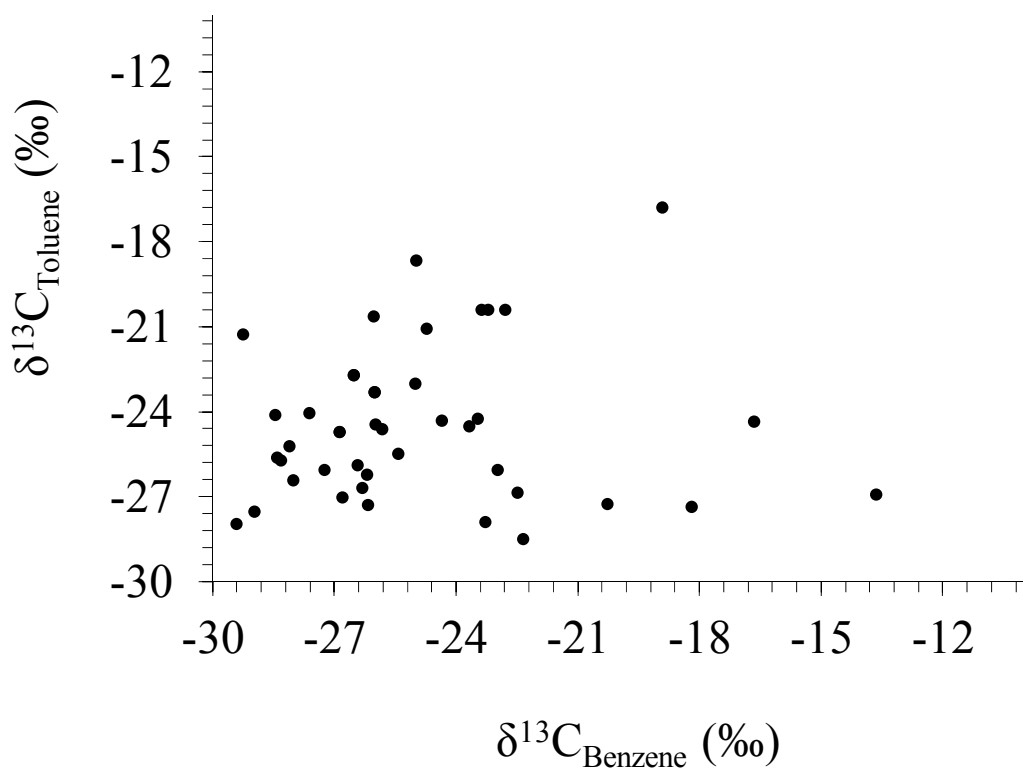

2    Figure 4





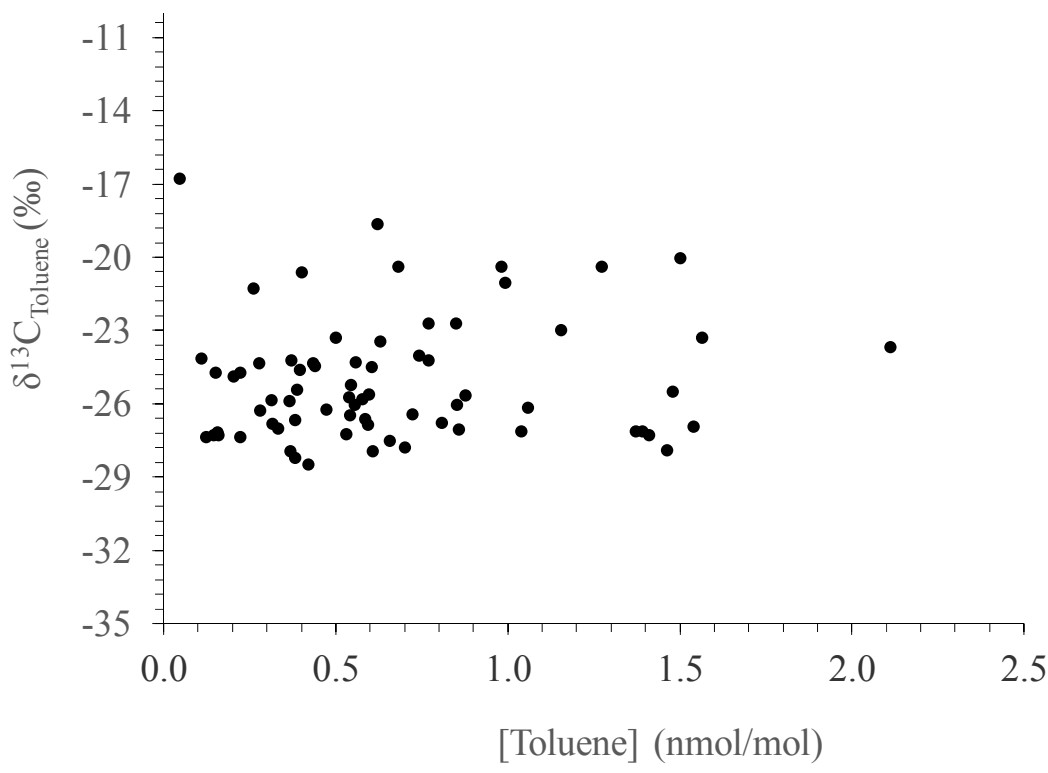

2     Figure 5



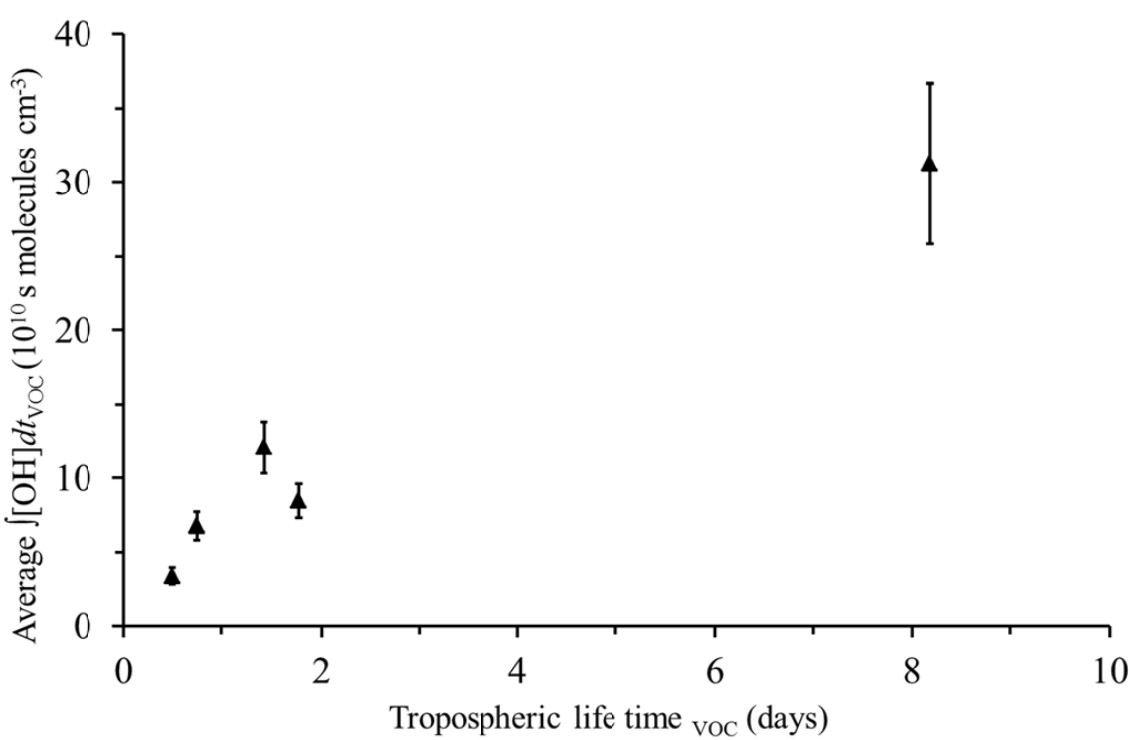

2      Figure 6





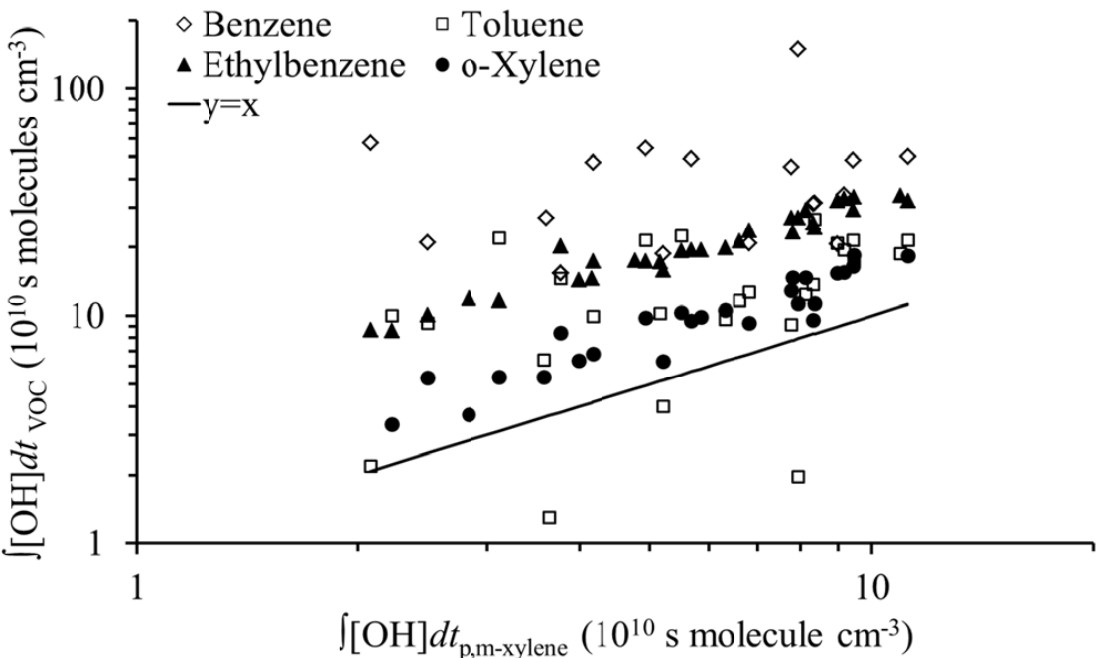

2    Figure 7



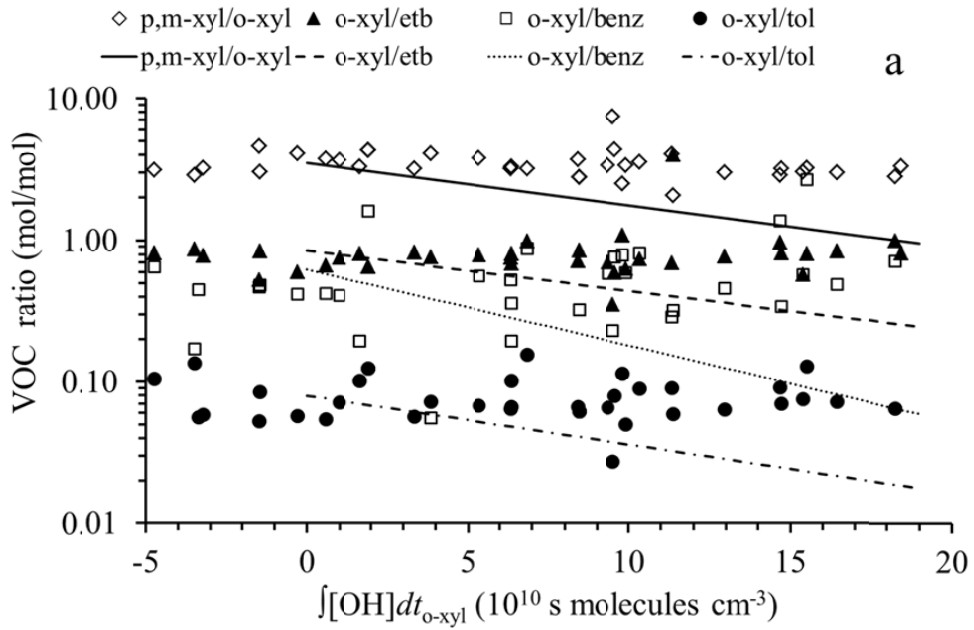

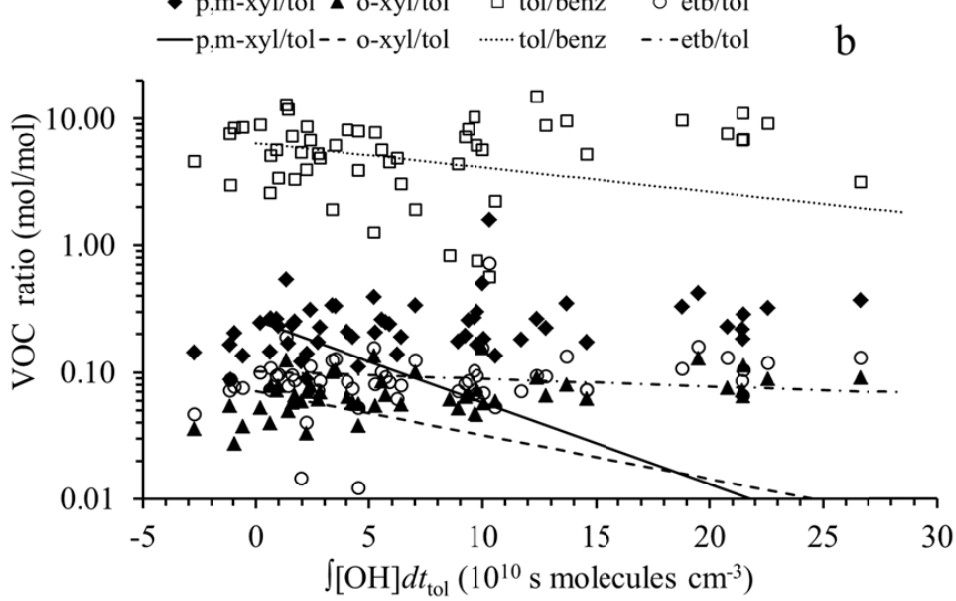

3    Figure 8





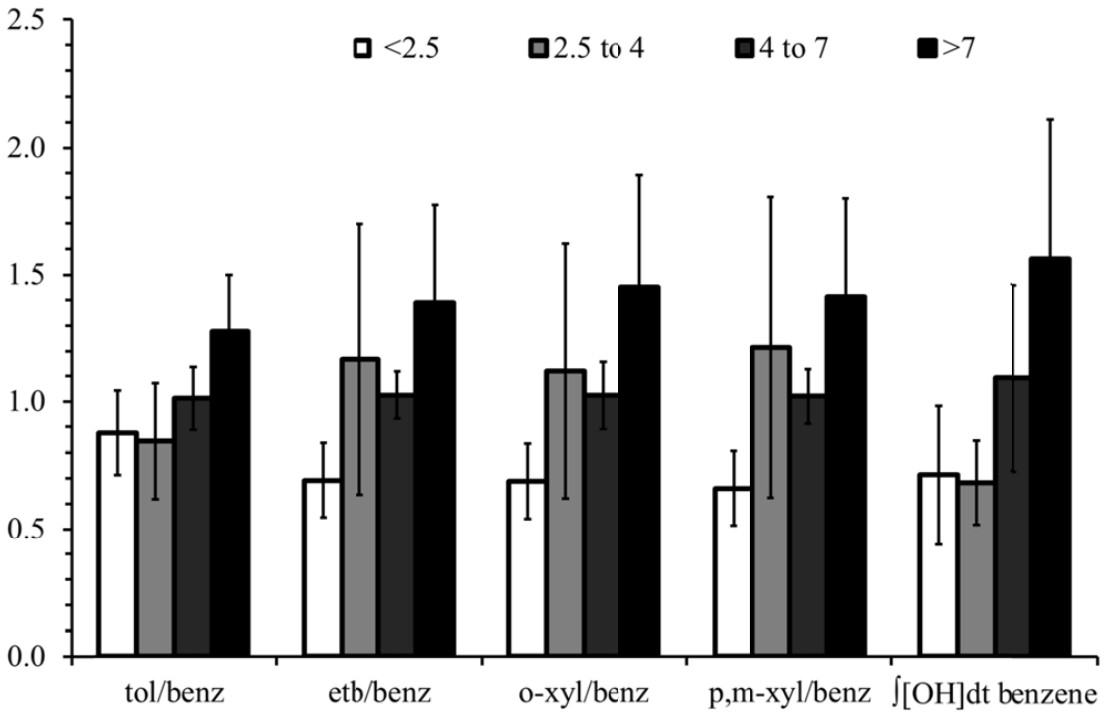

2      Figure 9





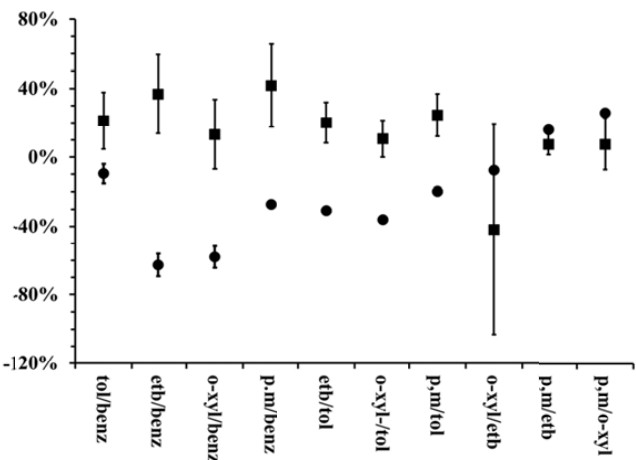

2 Figure 10



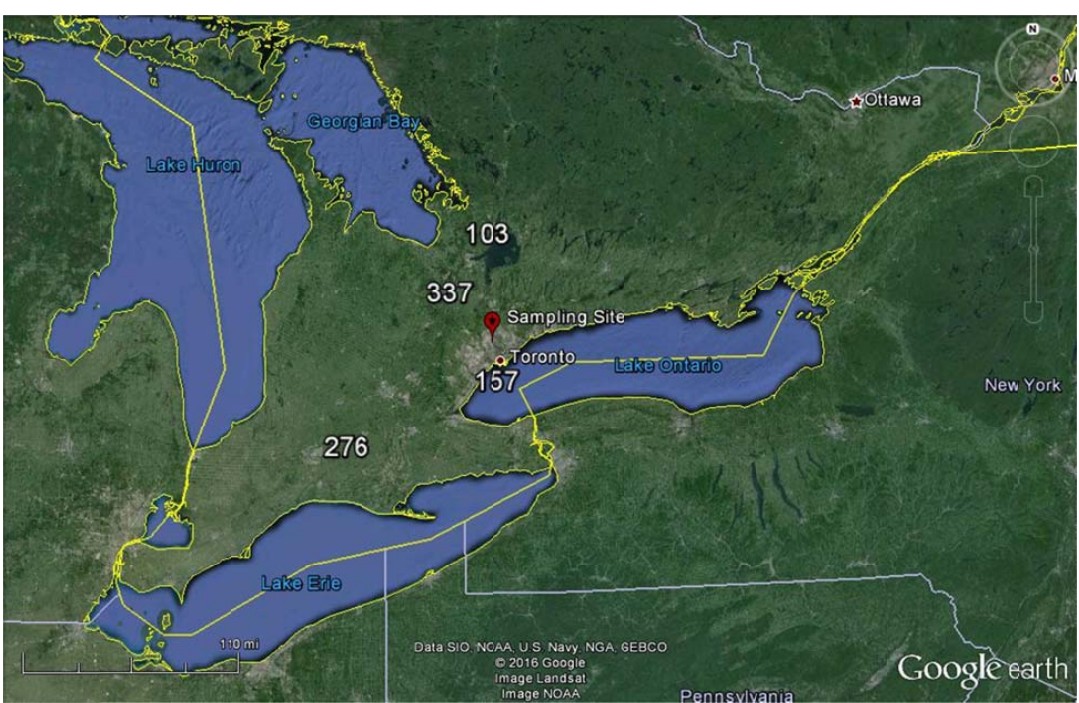

2    Figure 11



 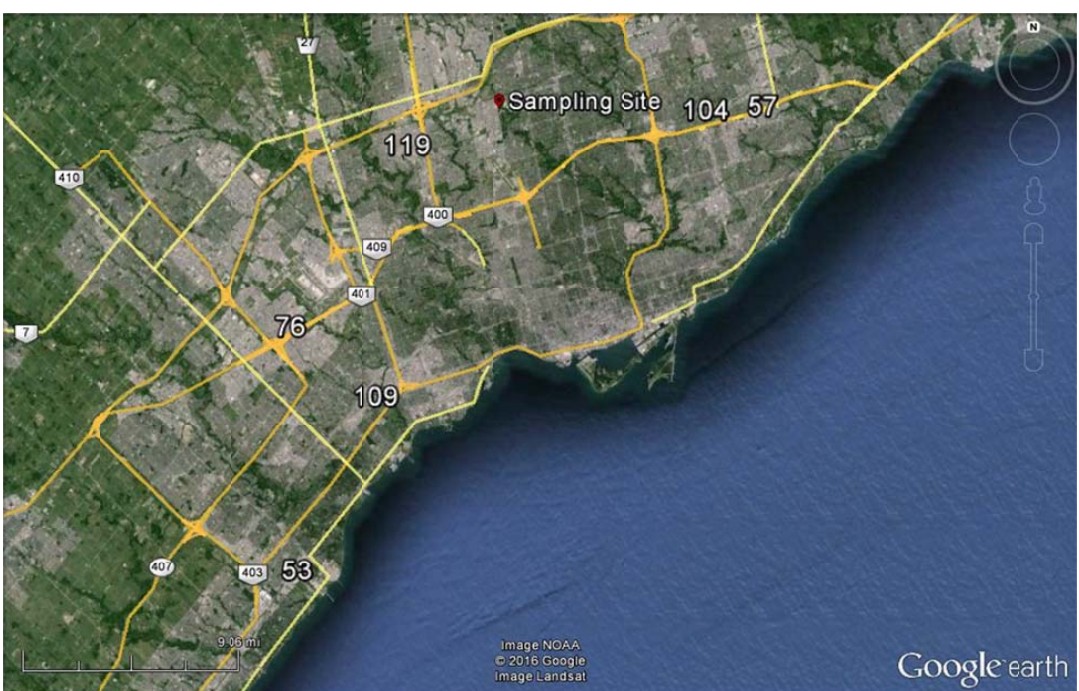

2    Figure 12