# Peer review of "Stable carbon isotope ratios of ambient aromatic volatile organic compounds"

_Atmospheric Chemistry and Physics, 2016_

## Author Comment (AC1) · 7 Jun 2016

[Figure]

Figure 2a

[Figure]

Figure 2b

[Figure]

Figure 2c

[Figure]

Figure 2d

[Figure]

Figure 2e

**Figure 2.** Frequency distribution for VOC carbon isotope ratios measured in Toronto in 2009 and 2010. Observations for different seasons are identified as follows: Fall- dotted, winter- black, and spring and summer- hatched. The solid vertical lines are the isotope composition of traffic related emissions (Rudolph et al., 2002).

[Figure]

Figure 3a

[Figure]

Figure 3b

[Figure]

Figure 3c

**Figure 3**. Example plots for the dependence between mixing ratios for different VOC pairs.

[Figure]

Figure 4a

[Figure]

Figure 4b

[Figure]

Figure 4c

**Figure 4**. Example plots for the dependence between carbon isotope ratios for different VOC pairs.

[Figure]

Figure 5a

[Figure]

Figure 5b.

**Figure 5**.Example plots of VOC mixing ratios versus carbon isotope ratios.

---

## Referee Comment (RC1) · Anonymous Referee #1 · 8 Jul 2016

Review on:

Stable carbon isotope ratios of ambient volatile organic compounds

This paper presents $\int[OH]dt$ obtained from stable carbon isotope ratios of ambient aromatic VOC measured at a suburban area in 2009 and 2010. The authors claim that the extent of photochemical processing of the different VOC depends strongly on the VOC reactivity. They show that for this study the differences in emission ratios are larger than the impact of photochemical aging.

The use of carbon isotope ratios for the study of atmospheric pollution and the chemistry of organic compounds in the atmosphere is a newly emerging tool. The experimental work and data interpretation are of high quality and excepting very few

points, that need to be addressed, both description and discussion of measurements are well founded. The manuscript contributes to scientific progress within the scope of the journal, therefore it is suitable to be published in ACP.

Some comments:
- Page8Line23-Page9Line2: reformulate. This information is not included in Fig. 2. Generally, the caption of Figure 2 must be more concrete. Is this for benzene, or for all VOC?
- Pages48-50: captions Fig. 3-5 in the main manuscript should contain the VOC of interest: 'Example plots ... VOC (benzene/toluene)...' (like in the supplement)
- Page9Lines5-7: reformulate. 'There is a strong dependence between...' and what?
- Page44Lines33-34: Sentence 'The bars represent the mean concentration ratios normalized relative to the average of all available ratios' needs some more explanation, at least in text. Generally, the axes of Fig.9-10 should be described.

Editorial revisions:
- Page5Line17: replace second '$\delta_A^{13}$C' by '$\delta_S^{13}$C'
- Page8Lines2and 22: replace 'Fig.2' by' Fig.3'
- Page8Line12: replace 'Figure 3' by' Figure 2'
- Page9Line14: replace 'Fig.3' by' Fig.2' and later in manuscript.
- Page40Table11: in third row insert a 'but' before $<\delta^{13}C_{global}$

---

## Referee Comment (RC2) · Anonymous Referee #2 · 25 Jul 2016

Review on:
Stable carbon isotope ratios of ambient volatile organic compounds
by Kornilova et al.

The authors present measurements of mixing ratios and stable carbon isotope ratios of aromatic compounds obtained in Toronto, Canada. They determined $\int [OH]dt$ to investigate the photochemical processing of the different VOC. Based on their results that $\int [OH]dt$ shows a dependence on VOC reactivity they arrive at the conclusion that mixing of air masses dominates the $\int [OH]dt$ values. Comparing the mixing ratios and isotope ratios of different pairs of VOC they conclude that sources with different emission ratios have a larger impact on the VOC composition than photochemical processing.

This study is yet another proof of the strength of isotope ratios for the investigation of atmospheric VOC. Although an interpretation maybe difficult (see my general comment), the attempt to disentangle transport and mixing from photochemical processes is an important approach to understand the organic composition of air masses and to identify sources and fate of these compounds.

The paper is the first investigating the relation between VOC concentrations and isotope ratios in an urban and more or less polluted environment. Therefore, it contributes to the progress of isotope ratios as a tool for the investigation of atmospheric processes that are not possible with concentration measurements alone. The paper should therefore be published in ACP after minor revision.

General comments:

I am sure, one can generate a scenario where two air masses containing emissions from two nearby sources or from a nearby and a distant source are mixed before reaching the sampling site leading to exactly the same emission ratios and isotope ratios because of different transport times, different emission ratios and different isotope ratios at emission. In this case you need another independent information (such as backward trajectories) to interpret your data, which generally can be done.

Here, the interpretation is based on measurement of samples with extremely long sampling times. How do these sampling times of 24 or even 64 hours affect the interpretation of the results? During these sampling times you may sample completely different air masses of different origin, transport times, emission ratios and source isotope ratios.

In the conclusion you state: ".. this does not allow differentiating between changes during sampling and mixing of air masses." I am not sure what you mean by "changes during sampling" (see below), changing emissions, changing wind directions? But nevertheless, don't you have to do this for your interpretation of the data?

Specific comments:

Title:
Change the title to "... ambient **aromatic** volatile organic compounds" because only these compounds are investigated (see first sentence of the abstract).

P 4, L 5-6:
"... mixing ratios that are 3 to 5 orders of magnitude lower ... "
With respect to which reference value? In the sentences before $\mu$mol mol$^{-1}$ and nmol mol$^{-1}$ are given. Maybe, giving mixing ratios here would be better.

P 4, L 11:
Reference: Eckstaedt et al, 2012
Please correct reference (also in the References):
The author is Christiane Vitzthum-von Eckstaedt: Vitzthum-von Eckstaedt, C. .....

P 4, equation 1 and P5, equation 2:
Skip "x 1000 ‰", see guidelines by Coplen 2011.

P 5, L 10 and L 17:
" ... is usually defined as the time integral of the OH concentration for an air mass ..."
In line 17 the "photochemical age" is mentioned. Using the isotope hydrocarbon clock approach, the photochemical age of the molecules of interest is determined. This is not necessarily the age of the air mass. Maybe, this should be mentioned here.

P 5, L17:
The second "$\delta_A{}^{13}C$" must read "$\delta_S{}^{13}C$".

P 7, 1. Paragraph
You used extremely long sampling times (24 h and even 64 h). Did you check the breakthrough volume of your sampling device? Can you comment on that?

P 8, Chapter 3.1
In this chapter references to all Figures have to be carefully checked.
Figure 3 is obviously a reference to Figure 2. In the Figure Caption to Figure 2 you should change "VOC" to "benzene".
P 8, L 22:
Figure 3 does not show the change of isotope ratios with season. Should there a Figure be added?
Figure 4 shows only benzene/toluene, Figure 5 shows only toluene, not all VOC

P 8, L 5:
"... strong dependence between $C_2$-alkylbenzenes ..."
Strong dependence of $C_2$-alkylbenzenes of what? Please specify.

P 9, L 14:
Figure 3 does not show isotope ratios (see above).

P 10, L 1:
Varying **the** extent of ...

P 10, L 6:
What do you mean by "steady state δ-values"?

P 10, L 19:
Again a wrong reference to Figure 3.

P 15, 2. Paragraph:

".. the observations are incompatible with the expected change of VOC concentration ratios"
What does that mean? Can that simply be explained by significantly different emission ratios?
Is there any evidence for that?

P 16, 1. Paragraph:
What could be the large difference in the sources? All compounds are anthropogenic
compounds emitted from similar sources with not too different emission ratios.

P 17, L 15:
Here a reference to Figure 15 is given regarding nearby sources. What are these sources? Can
they be specified?

P 21, L 15:
In the conclusion you state: ".. this does not allow differentiating between changes during
sampling and mixing of air masses." What do you mean by "changes during sampling"?

Table 3:
Some numbers are not consistent with Table 2.
In Table 2: Toluene $0.64 \pm 0.44$  -  in Table 3: Toluene $0.63 \pm 0.44$
In Table 2: Ethylbenzene: $0.07 \pm 0.05$ - in Table 3:  Ethylbenzene: $0.06 \pm 0.17$

Table 4:
Is there any possible explanation for the huge difference in the concentration ratios for
toluene/benzene and xylenes/benzene for the facilities Toronto? Is this due to differences
between automobile exhaust and evaporation losses? Is it possible to specify this?

References:
Please correct the reference to Niedojadlo et al.
The complete reference is:
... for NMVOCs, in: Simulation and Assessment of Chemical Processes in a Multiphase
Environment, NATO Science for Peace and Security, Series C: Environmental Security, ed.
by I. Barnes and M. M. Kharyatonov, Springer, the Netherlands, 2008.

Figure 9  and 10:
Labeling of the y-axes are missing

---

## Author Response (AR1)

We would like to thank both referees for their positive comments and the valuable suggestions for fixing some details, which will help us to prepare a revised manuscript meeting the high standards of ACP.
Both referees agree that isotope ratio measurements are potentially a very useful tool for investigating the atmospheric chemistry of volatile organic compounds (VOC), although referee 2 correctly points out that, unless additional information is available, there can be multiple solutions for explaining observed mixing ratios and carbon isotope ratios. (see below).

**Referee 1**
- Page8Line23-Page9Line2: reformulate. This information is not included in Fig. 2. Generally, the caption of Figure 2 must be more concrete. Is this for benzene, or for all VOC?
and
- Pages48-50: captions Fig. 3-5 in the main manuscript should contain the VOC of interest: 'Example plots ... VOC (benzene/toluene)...' (like in the supplement)

*In our revised version the examples from paper and supplement are combined in the paper to assure that the information mentioned in the text is consistent with the figures presented in the paper itself.*

- Page9Lines5-7: reformulate. 'There is a strong dependence between...' and what?
*Done: "There is a strong dependence between carbon isotope ratios of ethylbenzene and o-xylene, ethylbenzene and p,m-xylene as well as o-xylene and p,m-xylene. The correlations between the carbon isotope ratio of toluene and the $C_2$-alkylbenzenes are only weak and no correlation between the carbon isotope ratios of benzene and any of the other aromatic VOC is found (Fig. 4, Table 9).*

- Page44Lines33-34: Sentence 'The bars represent the mean concentration ratios normalized relative to the average of all available ratios' needs some more explanation, at least in text.
*Changed: (now page 45)"The bars represent the ratio of the mean concentration ratios for the chosen interval of ∫[OH]dt over the average for all concentration ratios."*

Generally, the axes of Fig.9-10 should be described.
*Axis labels added*

Editorial revisions:

- Page5Line17: replace second 'δ13A C' by 'δ13S C'
*Done*

- Page8Lines2and 22: replace 'Fig.2' by' Fig.3'
- Page8Line12: replace 'Figure 3' by' Figure 2'
- Page9Line14: replace 'Fig.3' by' Fig.2' and later in manuscript.
*Figure 2 and 3 have been switched in order to be consistent with the text and to make sure that the first mentioning of the figures is consistent with their numbering.*

- Page40Table11: in third row insert a 'but' before $<\delta13Cglobal$

*Done*

**Referee 2:**

General comments:

I am sure, one can generate a scenario where two air masses containing emissions from two nearby sources or from a nearby and a distant source are mixed before reaching the sampling site leading to exactly the same emission ratios and isotope ratios because of different transport times, different emission ratios and different isotope ratios at emission. In this case you need another independent information (such as backward trajectories) to interpret your data, which generally can be done.

*Reply: Meaningful constraints for such underdetermined systems are typically derived by including additional information, in this work we use the carbon isotope ratio of emissions, the emission ratios, and wind direction. It has to be understood that in a complex urban environment even with such additional information, there always will be a range of possible solutions. However, in many cases the use of isotope ratios can help to substantially reduce the range of valid solutions. In this study the information on photochemical VOC processing derived from isotope ratios eliminates the possibility that the observed VOC concentration ratios can be explained by photochemical aging of VOC emissions from known sources. The referee mentions the possible use of back trajectories for identification of VOC sources. In this work we decided to only use wind directions for several reasons. Based on the photochemical age determined from the carbon isotope ratios it can be concluded that in many cases the observed VOC levels are dominate by local (nearby) emissions. In this case wind direction provides the necessary information for filtering data sets based on impact from different point sources. The measurements were made in the greater Toronto area, an urban region with about five million inhabitants, dense traffic and a number of industrial facilities emitting substantial amounts of aromatic VOC. While back trajectories may allow a more detailed interpretation of individual measurements, filtering data sets by wind direction allows clear and easy testing for statistically meaningful differences.*

Here, the interpretation is based on measurement of samples with extremely long sampling times. How do these sampling times of 24 or even 64 hours affect the interpretation of the results? During these sampling times you may sample completely different air masses of different origin, transport times, emission ratios and source isotope ratios.

In the conclusion you state: ".. this does not allow differentiating between changes during sampling and mixing of air masses." I am not sure what you mean by "changes during sampling" (see below), changing emissions, changing wind directions? But nevertheless, don't you have to do this for your interpretation of the data?

*Reply: Indeed, this is a general problem for the interpretation of all measurements averaging over time intervals long enough to allow changes in wind direction, vertical mixing, emission rates etc. This problem is pointed out in our paper. In other words the averages do (obviously) not allow identification of changes with time within the sampling period. In the absence of published detailed data sets with higher time resolution it is not possible to differentiate between mixing of air masses and sampling air masses with different photochemical ages during the sampling period. However, this problem should not be confused with the fact that the observations still represent the correct*

*average values for the measurement intervals and can be interpreted as such.We modified our conclusions to clarify this point:*

*(P21, L22- P22,L5) This is useful for determining meaningful averages from a limited set of measurements, However, this does not allow differentiating between changes of ∫[OH]dt with time during the sampling period and mixing VOC with different ∫[OH]dt in the atmosphere prior to sampling. Separating the impact of the two processes on the VOC isotope ratio would require VOC isotope ratio measurements in samples collected during very short time periods of only a few minutes or less. To our knowledge no such VOC isotope ratio measurements in an urban environment have been published.*

Specific comments:
Title:
Change the title to "... ambient **aromatic** volatile organic compounds" because only these compounds are investigated (see first sentence of the abstract). 2
*Done*

P 4, L 5-6:
"... mixing ratios that are 3 to 5 orders of magnitude lower ... "
With respect to which reference value?
*Clarified: "mixing ratios that are 3 to 5 orders of magnitude lower than for methane or carbon monoxide,*
In the sentences before μmol mol-1 and nmol mol-1 are given. Maybe, giving mixing ratios here would be better.
*nmol mol$^{-1}$ etc. are IUPAC recommended units for mixing ratios*

P 4, L 11:
Reference: Eckstaedt et al, 2012
Please correct reference (also in the References):
The author is Christiane Vitzthum-von Eckstaedt: Vitzthum-von Eckstaedt, C. .....
*Done*

P 4, equation 1 and P5, equation 2:
Skip "x 1000 ‰", see guidelines by Coplen 2011.
*Literature is full of different ways to express delta values. In our opinion this is a matter of Journal style. The current form is based on a suggestion from the editor, if necessary we will make required changes in the final manuscript.*
P 5, L 10 and L 17:
" ... is usually defined as the time integral of the OH concentration for an air mass ..."
In line 17 the "photochemical age" is mentioned. Using the isotope hydrocarbon clock approach, the photochemical age of the molecules of interest is determined. This is not necessarily the age of the air mass. Maybe, this should be mentioned here.
*Reply: Indeed, it is one of the findings of this paper that different VOC have different PCAs and that it is therefore not correct to use the term PCA of an air mass. However, the "VOC ratio clock" assumes identical PCAs for different VOC, which in some literature is referred to as PCA of an air mass. We modified this part to avoid the impression of an inconsistency between introduction and discussion: "The term "photochemical age" is used for quantification of photochemical processing of a compound and it is usually defined as the time integral of the OH concentration (∫[OH]dt). Originally VOC concentration ratios were used to determine ∫[OH]dt, an approach that often is*

*referred to as hydrocarbon clock (Kornilova et al., 2015b; Stein and Rudolph, 2007; Kleinman et al., 2003; Rudolph et al., 2003; Thompson et al., 2003; Rudolph and Czuba, 2000; Jobson et al., 1999; Jobson et al., 1998). This concept implies that ∫[OH]dt is identical forthe VOC used for the calculation and sometimes the term photochemical history of an air masses is used (Parrish et al., 1992)."*

P 5, L17:

The second "$\delta_A{}^{13}C$" must read "$\delta_S{}^{13}C$".

*Done*

P 7, 1. Paragraph

You used extremely long sampling times (24 h and even 64 h). Did you check the breakthrough volume of your sampling device? Can you comment on that?

*Reply: In the cited paper by Kornilova et al. (which describes the measurement method) it is shown that (for the sampling conditions used) breakthrough does not occur for the studied VOC. We do not think that it is necessary to repeat details which are already published in a paper which describes and characterizes the used measurement method.*

P 8, Chapter 3.1

In this chapter references to all Figures have to be carefully checked.

Figure 3 is obviously a reference to Figure 2. In the Figure Caption to Figure 2 you should change "VOC" to "benzene".

P 8, L 22:

Figure 3 does not show the change of isotope ratios with season. Should there a Figure be added? Figure 4 shows only benzene/toluene, Figure 5 shows only toluene, not all VOC

*In our revised version the examples from paper and supplement are combined in the paper to assure that the information mentioned in the text is consistent with the figures presented in the paper itself. Figure 2 and 3 have been switched in order to be consistent with the text and to make sure that the first mentioning of the figures is consistent with their numbering.*

P 8, L 5:

"... strong dependence between C$_2$-alkylbenzenes ..."

Strong dependence of C$_2$-alkylbenzenes of what? Please specify.

*Changed: There is a strong dependence between carbon isotope ratios of ethylbenzene and o-xylene, ethylbenzene and p,m-xylene as well as o-xylene and p,m-xylene. The correlations between the carbon isotope ratio of toluene and the C$_2$-alkylbenzenes are only weak and no correlation between the carbon isotope ratios of benzene and any of the other aromatic VOC is found (Fig. 4, Table 9).*

P 9, L 14:

Figure 3 does not show isotope ratios (see above).

*Changed, see above*

P 10, L 1:

Varying **the** extent of ...

*Changed to: Variations in the extent of*

P 10, L 6:

What do you mean by "steady state ☐-values"?

*Clarified: Here $\delta^{13}C_{steady\ state}$ is the hypothetical average carbon isotope ratio of an atmospheric VOC in steady state between emissions and loss by reaction with the OH-radical.*

P 10, L 19:
Again a wrong reference to Figure 3.
*Fixed, see above*

P 15, 2. Paragraph:
".. the observations are incompatible with the expected change of VOC concentration ratios"
What does that mean? Can that simply be explained by significantly different emission ratios? Is there any evidence for that?
*Reply: Indeed, it is one of the conclusions drawn from the carbon isotope ratios that there must be sources with different emission ratios, which is discussed in more detail later in the paper.*
P 16, 1. Paragraph:
What could be the large difference in the sources? All compounds are anthropogenic compounds emitted from similar sources with not too different emission ratios.
*Reply: We are aware that there are quite a few publications, several of them cited in our paper, which base the interpretation of changes in VOC concentration ratios on this assumption. It is one of the conclusions drawn for the combination of isotope ratio and concentration ratio data that this assumption is (in our study) not justified. A more detailed look into sources with different emission ratios is presented later in the discussion.*

P 17, L 15:
Here a reference to Figure 15 is given regarding nearby sources. What are these sources? Can they be specified?
*Reply: Figure 12? The information is taken from the Canadian National Pollutant Release Inventory (as explained in the caption). This inventory is publicly available and cited. For the purpose of this paper the emission data are the relevant pieces of information. A presentation and discussion of the type of facilities and reasons for differences in emission ratios would not only add unnecessary length to our paper, but also distract from the main focus of the paper. Any reader interested in details can easily access the cited emission inventory and there are numerous publications discussing the factors that determine emission ratios.*

P 21, L 15:
In the conclusion you state: ".. this does not allow differentiating between changes during sampling and mixing of air masses." What do you mean by "changes during sampling"?
*Changed: "In this study sampling periods of at least several hours, in most cases of one day were used. This is useful for determining meaningful averages from a limited set of measurements, However, this does not allow differentiating between changes of ∫[OH]dt with time during the sampling period and mixing VOC with different ∫[OH]dt in the atmosphere prior to sampling. Separating the impact of the two processes on the VOC isotope ratio would require VOC isotope ratio measurements in samples collected during very short time periods of only a few minutes or less. To our knowledge no such VOC isotope ratio measurements in an urban environment have been published.*

Table 3:
Some numbers are not consistent with Table 2.
In Table 2: Toluene $0.64 \pm 0.44$ - in Table 3: Toluene $0.63 \pm 0.44$
In Table 2: Ethylbenzene: $0.07 \pm 0.05$ - in Table 3: Ethylbenzene: $0.06 \pm 0.17$
*Corrected*

Table 4:

Is there any possible explanation for the huge difference in the concentration ratios for toluene/benzene and xylenes/benzene for the facilities Toronto? Is this due to differences between automobile exhaust and evaporation losses? Is it possible to specify this?

*Reply: Table 14?: The Canadian National Pollutant Release Inventory provides emission data for (industrial) facilities, but no information on specific processes (or solvents and chemicals used). This information is rarely available (often proprietory). It can be speculated that for benzene, which is a known carcinogenic compound, emission controls in or close to densely populated areas are very tight. However, the important point is that our findings are consistent with the very low benzene emissions within the Toronto area. A discussion of processes and regulations behind emission data would be a paper with a very different focus.*

References:
Please correct the reference to Niedojadlo et al.
The complete reference is:

[revised manuscript text omitted]

Figure 2

[Figure]

[Figure]

[Figure]

[Figure]

[Figure]

Figure 3

[Figure]

[Figure]

[Figure]

Figure 4

[Figure]

[Figure]

Figure 5

[Figure]

Figure 6

[Figure]

Figure 7

[Figure]

[Figure]

Figure 8

[Figure]

Figure 9

[Figure]

Figure 10

[Figure]

Figure 11

[Figure]

Figure 12